# Correlated-PCA: Principal Components' Analysis when Data and Noise are Correlated

**Namrata Vaswani and Han Guo**
Iowa State University, Ames, IA, USA
Email: {namrata,hanguo}@iastate.edu

## Abstract

Given a matrix of observed data, Principal Components Analysis (PCA) computes a small number of orthogonal directions that contain most of its variability. Provably accurate solutions for PCA have been in use for a long time. However, to the best of our knowledge, all existing theoretical guarantees for it assume that the data and the corrupting noise are mutually independent, or at least uncorrelated. This is valid in practice often, but not always. In this paper, we study the PCA problem in the setting where the data and noise can be correlated. Such noise is often also referred to as "data-dependent noise". We obtain a correctness result for the standard eigenvalue decomposition (EVD) based solution to PCA under simple assumptions on the data-noise correlation. We also develop and analyze a generalization of EVD, cluster-EVD, that improves upon EVD in certain regimes.

## 1   Introduction

Principal Components Analysis (PCA) is among the most frequently used tools for dimension reduction. Given a matrix of data, it computes a small number of orthogonal directions that contain all (or most) of the variability of the data. The subspace spanned by these directions is the "principal subspace". To use PCA for dimension reduction, one projects the observed data onto this subspace. The standard solution to PCA is to compute the reduced singular value decomposition (SVD) of the data matrix, or, equivalently, to compute the reduced eigenvalue decomposition (EVD) of the empirical covariance matrix of the data. If all eigenvalues are nonzero, a threshold is used and all eigenvectors with eigenvalues above the threshold are retained. This solution, which we henceforth refer to as *simple EVD*, or just *EVD*, has been used for many decades and is well-studied in literature, e.g., see [1] and references therein. However, to the best of our knowledge, all existing results for it assume that the true data and the corrupting noise in the observed data are independent, or, at least, uncorrelated. This is valid in practice often, but not always. Here, we study the PCA problem in the setting where the data and noise vectors may be correlated (correlated-PCA). Such noise is sometimes called "data-dependent" noise.

**Contributions.**   (1) Under a boundedness assumption on the true data vectors, and some other assumptions, for a fixed desired subspace error level, we show that the sample complexity of simple-EVD for correlated-PCA scales as $f^2 r^2 \log n$ where $n$ is the data vector length, $f$ is the condition number of the true data covariance matrix and $r$ is its rank. Here "sample complexity" refers to the number of samples needed to get a small enough subspace recovery error with high probability (whp). The dependence on $f^2$ is problematic for datasets with large condition numbers, and, especially in the high dimensional setting where $n$ is large. (2) To address this, we also develop and analyze a generalization of simple-EVD, called *cluster-EVD*. Under an eigenvalues' "clustering" assumption, cluster-EVD weakens the dependence on $f$.

To our best knowledge, the correlated-PCA problem has not been explicitly studied. We first encountered it while solving the dynamic robust PCA problem in the Recursive Projected Compressive Sensing (ReProCS) framework [2, 3, 4, 5]. The version of correlated-PCA studied here is motivated

by these works. Some other somewhat related recent works include [6, 7] that study stochastic optimization based techniques for PCA; and [8, 9, 10, 11] that study online PCA.

**Notation.** We use the interval notation $[a, b]$ to mean all of the integers between $a$ and $b$, inclusive, and similarly for $[a, b)$ etc. We use $\| \cdot \|$ to denote the $l_2$ norm of a vector or the induced $l_2$ norm of a matrix. For other $l_p$ norms, we use $\| \cdot \|_p$. For a set $\mathcal{T}$, $\boldsymbol{I}_{\mathcal{T}}$ refers to an $n \times |\mathcal{T}|$ matrix of columns of the identity matrix indexed by entries in $\mathcal{T}$. For a matrix $\boldsymbol{A}$, $\boldsymbol{A}_{\mathcal{T}} := \boldsymbol{A}\boldsymbol{I}_{\mathcal{T}}$. A tall matrix with orthonormal columns is referred to as a *basis matrix*. For basis matrices $\hat{\boldsymbol{P}}$ and $\boldsymbol{P}$, we quantify the subspace error (SE) between their range spaces using

$$\mathrm{SE}(\hat{\boldsymbol{P}}, \boldsymbol{P}) := \|(\boldsymbol{I} - \hat{\boldsymbol{P}}\hat{\boldsymbol{P}}')\boldsymbol{P}\|. \tag{1}$$

## 1.1 Correlated-PCA: Problem Definition

We are given a time sequence of data vectors, $\boldsymbol{y}_t$, that satisfy

$$\boldsymbol{y}_t = \boldsymbol{\ell}_t + \boldsymbol{w}_t, \text{ with } \boldsymbol{w}_t = \boldsymbol{M}_t \boldsymbol{\ell}_t \text{ and } \boldsymbol{\ell}_t = \boldsymbol{P}\boldsymbol{a}_t \tag{2}$$

where $\boldsymbol{P}$ is an $n \times r$ basis matrix with $r \ll n$. Here $\boldsymbol{\ell}_t$ is the true data vector that lies in a low dimensional subspace of $\mathbb{R}^n$, $\mathrm{range}(\boldsymbol{P})$; $\boldsymbol{a}_t$ is its projection into this $r$-dimensional subspace; and $\boldsymbol{w}_t$ is the data-dependent noise. We refer to $\boldsymbol{M}_t$ as the correlation / data-dependency matrix. The *goal* is to estimate $\mathrm{range}(\boldsymbol{P})$. We make the following assumptions on $\boldsymbol{\ell}_t$ and $\boldsymbol{M}_t$.

**Assumption 1.1.** *The subspace projection coefficients, $\boldsymbol{a}_t$, are zero mean, mutually independent and bounded random vectors (r.v.), with a diagonal covariance matrix $\boldsymbol{\Lambda}$. Define $\lambda^- := \lambda_{\min}(\boldsymbol{\Lambda})$, $\lambda^+ := \lambda_{\max}(\boldsymbol{\Lambda})$ and $f := \frac{\lambda^+}{\lambda^-}$. Since the $\boldsymbol{a}_t$'s are bounded, we can also define a finite constant $\eta := \max_{j=1,2,\dots r} \max_t \frac{(\boldsymbol{a}_t)_j^2}{\lambda_j}$. Thus, $(\boldsymbol{a}_t)_j^2 \le \eta \lambda_j$.*

For most bounded distributions, $\eta$ will be a small constant more than one, e.g., if the distribution of all entries of $\boldsymbol{a}_t$ is iid zero mean uniform, then $\eta = 3$. From Assumption 1.1, clearly, the $\boldsymbol{\ell}_t$'s are also zero mean, bounded, and mutually independent r.v.'s with a rank $r$ covariance matrix $\boldsymbol{\Sigma} \overset{\mathrm{EVD}}{=} \boldsymbol{P}\boldsymbol{\Lambda}\boldsymbol{P}'$. In the model, for simplicity, we assume $\boldsymbol{\Lambda}$ to be fixed. However, even if we replace $\boldsymbol{\Lambda}$ by $\boldsymbol{\Lambda}_t$ and define $\lambda^- = \min_t \lambda_{\min}(\boldsymbol{\Lambda}_t)$ and $\lambda^+ = \lambda_{\max}(\boldsymbol{\Lambda}_t)$, all our results will still hold.

**Assumption 1.2.** *Decompose $\boldsymbol{M}_t$ as $\boldsymbol{M}_t = \boldsymbol{M}_{2,t}\boldsymbol{M}_{1,t}$. Assume that*

$$\|\boldsymbol{M}_{1,t}\boldsymbol{P}\| \le q < 1, \ \|\boldsymbol{M}_{2,t}\| \le 1, \tag{3}$$

*and, for any sequence of positive semi-definite Hermitian matrices, $\boldsymbol{A}_t$, the following holds*

$$\text{for a } \beta < \alpha, \ \left\| \frac{1}{\alpha} \sum_{t=1}^{\alpha} \boldsymbol{M}_{2,t} \boldsymbol{A}_t \boldsymbol{M}_{2,t}' \right\| \le \frac{\beta}{\alpha} \max_{t \in [1,\alpha]} \|\boldsymbol{A}_t\|. \tag{4}$$

We will need the above to hold for all $\alpha \ge \alpha_0$ and for all $\beta \le c_0 \alpha$ with a $c_0 \ll 1$. We set $\alpha_0$ and $c_0$ in Theorems 2.1 and 3.3; both will depend on $q$. Observe that, using (3), $\frac{\|\boldsymbol{w}_t\|}{\|\boldsymbol{\ell}_t\|} \le q$, and so $q$ *is an upper bound on the signal-to-noise ratio (SNR).*

To understand the assumption on $\boldsymbol{M}_{2,t}$, notice that, if we allow $\beta = \alpha$, then (4) always holds and is not an assumption. Let $\boldsymbol{B}$ denote the matrix on the LHS of (4). One example situation when (4) holds with a $\beta \ll \alpha$ is if $\boldsymbol{B}$ is block-diagonal with blocks $\boldsymbol{A}_t$. In this case, it holds with $\beta = 1$. In fact, it also holds with $\beta = 1$ if $\boldsymbol{B}$ is permutation-similar to a block diagonal matrix. The matrix $\boldsymbol{B}$ will be of this form if $\boldsymbol{M}_{2,t} = \boldsymbol{I}_{\mathcal{T}_t}$ with all the sets $\mathcal{T}_t$ being mutually disjoint. More generally, if $\boldsymbol{B}$ is permutation-similar to a block-diagonal matrix with blocks given by the summation of $\boldsymbol{A}_t$'s over at most $\beta_0 < \alpha$ time instants, then (4) holds with $\beta = \beta_0$. This will happen if $\boldsymbol{M}_{2,t} = \boldsymbol{I}_{\mathcal{T}_t}$ with $\mathcal{T}_t = \mathcal{T}^{[k]}$ for at most $\beta_0$ time instants and if sets $\mathcal{T}^{[k]}$ are mutually disjoint for different $k$. Finally, the $\mathcal{T}^{[k]}$'s need not even be mutually disjoint. As long as they are such that $\boldsymbol{B}$ is a matrix with nonzero blocks on only the main diagonal and on a few diagonals near it, e.g., if it is block tri-diagonal, it can be shown that the above assumption holds. This example is generalized in Assumption 1.3 given below.

## 1.2 Examples of correlated-PCA problems

One key example of correlated-PCA is the *PCA with missing data (PCA-missing)* problem. Let $\mathcal{T}_t$ denote the set of missing entries at time $t$. Suppose, we set the missing entries of $\boldsymbol{y}_t$ to zero. Then,

$$\boldsymbol{y}_t = \boldsymbol{\ell}_t - \boldsymbol{I}_{\mathcal{T}_t}\boldsymbol{I}_{\mathcal{T}_t}'\boldsymbol{\ell}_t. \tag{5}$$

In this case $M_{2,t} = I_{\mathcal{T}_t}$ and $M_{1,t} = -I_{\mathcal{T}_t}{}'$. Thus, $q$ is an upper bound on $\|I_{\mathcal{T}_t}{}'P\|$. Clearly, it will be small if the columns of $P$ are dense vectors. For the reader familiar with low-rank matrix completion (MC), e.g., [12, 13], *PCA-missing* can also be solved by first solving the low-rank matrix completion problem to recover $L$, followed by PCA on the completed matrix. This would, of course, be much more expensive than directly solving *PCA-missing* and would need more assumptions.

Another example where correlated-PCA occurs is that of robust PCA (low-rank + sparse formulation) [14, 15, 16] when the sparse component's magnitude is correlated with $\ell_t$. Let $\mathcal{T}_t$ denote the support set of $w_t$ and let $x_t$ be the $|\mathcal{T}_t|$-length vector of its nonzero entries. If we assume linear dependency of $x_t$ on $\ell_t$, we can write out $y_t$ as

$$y_t = \ell_t + I_{\mathcal{T}_t} x_t = \ell_t + I_{\mathcal{T}_t} M_{s,t} \ell_t. \tag{6}$$

Thus $M_{2,t} = I_{\mathcal{T}_t}$ and $M_{1,t} = M_{s,t}$ and so $q$ is an upper bound on $\|M_{s,t}P\|$. In the rest of the paper, we refer to this problem is *"PCA with sparse data-dependent corruptions (PCA-SDDC)"*. One key application where it occurs is in foreground-background separation for videos consisting of a slow changing background sequence (modeled as lying close to a low-dimensional subspace) and a sparse foreground image sequence consisting typically of one or more moving objects [14]. The PCA-SDDC problem is to estimate the background sequence's subspace. In this case, $\ell_t$ is the background image at time $t$, $\mathcal{T}_t$ is the support set of the foreground image at $t$, and $x_t$ is the difference between foreground and background intensities on $\mathcal{T}_t$. An alternative solution approach for PCA-SDDC is to use an RPCA solution such as principal components' pursuit (PCP) [14, 15] or Alternating-Minimization (Alt-Min-RPCA) [17] to first recover the matrix $L$ followed by PCA on $L$. However, as shown in Sec. 5, Table 1, this approach will be much slower; and it will work only if its required incoherence assumptions hold. For example, if the columns of $P$ are sparse, it fails.

For both problems above, a solution for PCA will work only when the *corrupting noise $w_t$ is small compared to $\ell_t$*. A sufficient condition for this is that $q$ is small.

A third example where correlated-PCA and its generalization, correlated-PCA with partial subspace knowledge, occurs is in the subspace update step of Recursive Projected Compressive Sensing (Re-ProCS) for dynamic robust PCA [3, 5].

In all three of the above applications, the assumptions on the data-noise correlation matrix given in Assumption 1.2 hold if there are enough changes of a certain type in the set of missing or corrupted entries, $\mathcal{T}_t$. One example where this is true is in case of a 1D object of length $s$ or less that remains static for at most $\beta$ frames at a time. When it moves, it moves by at least a certain fraction of $s$ pixels. The following assumption is inspired by the object's support.

**Assumption 1.3.** *Let $l$ denote the number of times the set $\mathcal{T}_t$ changes in the interval $[1, \alpha]$ (or in any given interval of length $\alpha$ in case of dynamic robust PCA). So $0 \leq l \leq \alpha - 1$. Let $t^0 := 1$; let $t^k$, with $t^k < t^{k+1}$, denote the time instants in this interval at which $\mathcal{T}_t$ changes; and let $\mathcal{T}^{[k]}$ denote the distinct sets. In other words, $\mathcal{T}_t = \mathcal{T}^{[k]}$ for $t \in [t^k, t^{k+1})$, for each $k = 1, 2, \ldots, l$. Assume that the following hold with a $\beta < \alpha$:*

1. *$(t^{k+1} - t^k) \leq \tilde{\beta}$ and $|\mathcal{T}^{[k]}| \leq s$;*

2. *$\rho^2 \tilde{\beta} \leq \beta$ where $\rho$ is the smallest positive integer so that, for any $0 \leq k \leq l$, $\mathcal{T}^{[k]}$ and $\mathcal{T}^{[k+\rho]}$ are disjoint;*

3. *for any $k_1, k_2$ satisfying $0 \leq k_1 < k_2 \leq l$, the sets $(\mathcal{T}^{[k_1]} \setminus \mathcal{T}^{[k_1+1]})$ and $(\mathcal{T}^{[k_2]} \setminus \mathcal{T}^{[k_2+1]})$ are disjoint.*

*An implicit assumption for condition 3 to hold is that $\sum_{k=0}^{l} |\mathcal{T}^{[k]} \setminus \mathcal{T}^{[k+1]}| \leq n$. Observe that conditions 2 and 3 enforce an upper bound on the maximum support size $s$.*

To connect Assumption 1.3 with the moving object example given above, condition 1 holds if the object's size is at most $s$ and if it moves at least once every $\tilde{\beta}$ frames. Condition 2 holds, if, every time it moves, it moves in the same direction and by at least $\frac{s}{\rho}$ pixels. Condition 3 holds if, every time it moves, it moves in the same direction and by at most $d_0 \geq \frac{s}{\rho}$ pixels, with $d_0 \alpha \leq n$ (or, more generally, the motion is such that, if the object were to move at each frame, and if it started at the top of the frame, it does not reach the bottom of the frame in a time interval of length $\alpha$).

The following lemma [4] shows that, with Assumption 1.3 on $\mathcal{T}_t$, $M_{2,t} = I_{\mathcal{T}_t}$ satisfies the assumption on $M_{2,t}$ given in Assumption 1.2. Its proof generalizes the discussion below Assumption 1.2.

**Lemma 1.4.** *[[4], Lemmas 5.2 and 5.3] Assume that Assumption 1.3 holds. For any sequence of $|\mathcal{T}_t| \times |\mathcal{T}_t|$ symmetric positive-semi-definite matrices $\boldsymbol{A}_t$,*

$$\| \sum_{t=1}^{\alpha} \boldsymbol{I}_{\mathcal{T}_t} \boldsymbol{A}_t \boldsymbol{I}_{\mathcal{T}_t}{}' \| \leq (\rho^2 \tilde{\beta}) \max_{t \in [1,\alpha]} \|\boldsymbol{A}_t\| \leq \beta \max_{t \in [1,\alpha]} \|\boldsymbol{A}_t\|$$

*Thus, if $\|\boldsymbol{I}_{\mathcal{T}_t}{}'\boldsymbol{P}\| \leq q < 1$, then the PCA-missing problem satisfies Assumption 1.2. If $\|\boldsymbol{M}_{s,t}\boldsymbol{P}\| \leq q < 1$, then the PCA-SDDC problem satisfies Assumption 1.2.*

Assumption 1.3 is one model on $\mathcal{T}_t$ that ensures that, if $\boldsymbol{M}_{2,t} = \boldsymbol{I}_{\mathcal{T}_t}$, the assumption on $\boldsymbol{M}_{2,t}$ given in Assumption 1.2 holds. For its many generalizations, see Supplementary Material, Sec. 7, or [4].

As explained in [18], data-dependent noise also often occurs in molecular biology applications when the noise affects the measurement levels through the very same process as the interesting signal.

## 2   Simple EVD

Simple EVD computes the top eigenvectors of the empirical covariance matrix, $\frac{1}{\alpha} \sum_{t=1}^{\alpha} \boldsymbol{y}_t \boldsymbol{y}_t{}'$, of the observed data. The following can be shown.

**Theorem 2.1** (simple-EVD result). *Let $\hat{\boldsymbol{P}}$ denote the matrix containing all the eigenvectors of $\frac{1}{\alpha} \sum_{t=1}^{\alpha} \boldsymbol{y}_t \boldsymbol{y}_t{}'$ with eigenvalues above a threshold, $\lambda_{\mathrm{thresh}}$, as its columns. Pick a $\zeta$ so that $r\zeta \leq 0.01$. Suppose that $\boldsymbol{y}_t$'s satisfy (2) and the following hold.*

1. *Assumption 1.1 on $\boldsymbol{\ell}_t$ holds. Define*

$$\alpha_0 := C\eta^2 \frac{r^2 11 \log n}{(r\zeta)^2} \max(f, qf, q^2 f)^2, \; C := \frac{32}{0.01^2}.$$

2. *Assumption 1.2 on $\boldsymbol{M}_t$ holds for any $\alpha \geq \alpha_0$ and for any $\beta$ satisfying*

$$\frac{\beta}{\alpha} \leq \left( \frac{1 - r\zeta}{2} \right)^2 \min\left( \frac{(r\zeta)^2}{4.1(qf)^2}, \frac{(r\zeta)}{q^2 f} \right)$$

3. *Set algorithm parameters $\lambda_{\mathrm{thresh}} = 0.95\lambda^-$ and $\alpha \geq \alpha_0$.*

*Then, with probability at least $1 - 6n^{-10}$, $\mathrm{SE}(\hat{\boldsymbol{P}}, \boldsymbol{P}) \leq r\zeta$.*

*Proof:*  The proof involves a careful application of the $\sin\theta$ theorem [19] to bound the subspace error, followed by using matrix Hoeffding [20] to obtain high probability bounds on each of the terms in the $\sin\theta$ bound. It is given in the Supplementary Material, Section 8.

Consider the lower bound on $\alpha$. We refer to this as the "sample complexity". Since $q < 1$, and $\eta$ is a small constant (e.g., for the uniform distribution, $\eta = 3$), for a fixed error level, $r\zeta$, $\alpha_0$ simplifies to $cf^2r^2 \log n$. Notice that the dependence on $n$ is logarithmic. It is possible to show that the sample complexity scales as $\log n$ because we assume that the $\boldsymbol{\ell}_t$'s are bounded r.v.s. As a result we can apply the matrix Hoeffding inequality [20] to bound the perturbation between the observed data's empirical covariance matrix and that of the true data. The bounded r.v. assumption is actually a more practical one than the usual Gaussian assumption since most sources of data have finite power.

By replacing matrix Hoeffding by Theorem 5.39 of [21] in places where one can apply a concentration of measure result to $\sum_t \boldsymbol{a}_t \boldsymbol{a}_t{}'/\alpha$ (which is at $r \times r$ matrix), and by matrix Bernstein [20] elsewhere, it should be possible to further reduce the sample complexity to $c \max((qf)^2 r \log n, f^2(r + \log n))$. It should also be possible remove the boundedness assumption and replace it by a Gaussian (or a sub-Gaussian) assumption, but, that would increase the sample complexity to $c(qf)^2 n$.

Consider the upper bound on $\beta/\alpha$. Clearly, the smaller term is the first one. This depends on $1/(qf)^2$. Thus, when $f$ is large and $q$ is not small enough, the bound required may be impractically small. As will be evident from the proof (see Remark 8.3 in Supplementary Material), we get this bound because $\boldsymbol{w}_t$ is correlated with $\boldsymbol{\ell}_t$ and this results in $\mathbb{E}[\boldsymbol{\ell}_t \boldsymbol{w}_t{}'] \neq 0$.

If $\boldsymbol{w}_t$ and $\boldsymbol{\ell}_t$ were uncorrelated, $qf$ would get replaced by $\frac{\lambda_{\max}(\mathrm{Cov}(\boldsymbol{w}_t))}{\lambda^-}$ in the upper bound on $\beta/\alpha$ as well as in the sample complexity.

**Application to PCA-missing and PCA-SDDC.** By Lemma 1.4, the following is immediate.

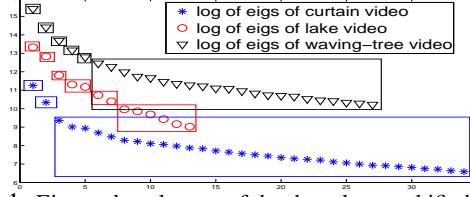

Figure 1: Eigenvalue clusters of the three low-rankified videos.

**Corollary 2.2.** *Consider the PCA-missing model, (5), and assume that $\max_t \|\boldsymbol{I}_{\mathcal{T}_t}'\boldsymbol{P}\| \leq q < 1$; or consider the PCA-SDDC model, (6), and assume that $\max_t \|\boldsymbol{M}_{s,t}\boldsymbol{P}\| \leq q < 1$. Assume that everything in Theorem 2.1 holds except that we replace Assumption 1.2 by Assumption 1.3. Then, with probability at least $1 - 6n^{-10}$, $\mathrm{SE}(\hat{\boldsymbol{P}}, \boldsymbol{P}) \leq r\zeta$.*

## 3 Cluster-EVD

To try to relax the strong dependence on $f^2$ of the result above, we develop a generalization of simple-EVD that we call *cluster-EVD*. This requires the clustering assumption.

### 3.1 Clustering assumption

To state the assumption, define the following partition of the index set $\{1, 2, \ldots r\}$ based on the eigenvalues of $\boldsymbol{\Sigma}$. Let $\lambda_i$ denote its $i$-th largest eigenvalue.

**Definition 3.1** ($g$-condition-number partition of $\{1, 2, \ldots r\}$)**.** *Define $\mathcal{G}_1 = \{1, 2, \ldots r_1\}$ where $r_1$ is the index for which $\frac{\lambda_1}{\lambda_{r_1}} \leq g$ and $\frac{\lambda_1}{\lambda_{r_1+1}} > g$. In words, to define $\mathcal{G}_1$, start with the index of the first (largest) eigenvalue and keep adding indices of the smaller eigenvalues to the set until the ratio of the maximum to the minimum eigenvalue first exceeds $g$.*

*For each $k > 1$, define $\mathcal{G}_k = \{r_* + 1, r_* + 2, \ldots, r_* + r_k\}$ where $r_* = (\sum_{i=1}^{k-1} r_i)$, $r_k$ is the index for which $\frac{\lambda_{r_*+1}}{\lambda_{r_*+r_k}} \leq g$ and $\frac{\lambda_{r_*+1}}{\lambda_{r_*+r_k+1}} > g$. In words, to define $\mathcal{G}_k$, start with the index of the $(r_* + 1)$-th eigenvalue, and repeat the above procedure.*

*Stop when $\lambda_{r_*+r_k+1} = 0$, i.e., when there are no more nonzero eigenvalues. Define $\vartheta = k$ as the number of sets in the partition. Thus $\{\mathcal{G}_1, \mathcal{G}_2, \ldots, \mathcal{G}_\vartheta\}$ is the desired partition.*

Define $\boldsymbol{G}_0 = [.]$, $\boldsymbol{G}_k := (\boldsymbol{P})_{\mathcal{G}_k}$, $\lambda_k^+ := \max_{i \in \mathcal{G}_k} \lambda_i(\boldsymbol{\Lambda})$, $\lambda_k^- := \min_{i \in \mathcal{G}_k} \lambda_i(\boldsymbol{\Lambda})$ and

$$\chi := \max_{k=1,2,\ldots,\vartheta} \frac{\lambda_{k+1}^+}{\lambda_k^-}.$$

$\chi$ quantifies the "distance" between consecutive sets of the above partition. Moreover, by definition, $\frac{\lambda_k^+}{\lambda_k^-} \leq g$. Clearly, $g \geq 1$ and $\chi \leq 1$ always. We assume the following.

**Assumption 3.2.** *For a $1 \leq g^+ < f$ and a $0 \leq \chi^+ < 1$, assume that there exists a $g$ satisfying $1 \leq g \leq g^+$ and a $\chi$ satisfying $0 \leq \chi \leq \chi^+$, for which we can define a $g$-condition-number partition of $\{1, 2, \ldots r\}$ that satisfies $\chi \leq \chi^+$. The number of sets in the partition is $\vartheta$. When $g^+$ and $\chi^+$ are small, we say that the eigenvalues are "well-clustered" with "clusters", $\mathcal{G}_k$.*

This assumption can be understood as a generalization of the eigen-gap condition needed by the block power method, which is a fast algorithm for obtaining the $k$ top eigenvectors of a matrix [22]. We expect it to hold for data that has variability across different scales. The large scale variations would result in the first (largest eigenvalues') cluster and the smaller scale variations would form the later clusters. This would be true, for example, for video "textures" such as moving waters or waving trees in a forest. We tested this assumption on some such videos. We describe our conclusions here for three videos - "lake" (video of moving lake waters), "waving-tree" (video consisting of waving trees), and "curtain" (video of window curtains moving due to the wind). For each video, we first made it low-rank by keeping the eigenvectors corresponding to the smallest number of eigenvalues that contain at least 90% of the total energy and projecting the video onto this subspace. For the low-rankified lake video, $f = 74$ and Assumption 3.2 holds with $\vartheta = 6$ clusters, $g^+ = 2.6$ and $\chi^+ = 0.7$. For the waving-tree video, $f = 180$ and Assumption 3.2 holds with $\vartheta = 6$, $g^+ = 9.4$ and $\chi^+ = 0.72$. For the curtain video, $f = 107$ and the assumption holds $\vartheta = 3$, $g^+ = 16.1$ and $\chi^+ = 0.5$. We show the clusters of eigenvalues in Fig. 1.

---

**Algorithm 1 Cluster-EVD**

**Parameters:** $\alpha, \hat{g}, \lambda_{\text{thresh}}$.
Set $\hat{\boldsymbol{G}}_0 \leftarrow [.]$. Set the flag $\text{Stop} \leftarrow 0$. Set $k \leftarrow 1$.
Repeat

1. Let $\hat{\boldsymbol{G}}_{\text{det},k} := [\hat{\boldsymbol{G}}_0, \hat{\boldsymbol{G}}_1, \ldots \hat{\boldsymbol{G}}_{k-1}]$ and let $\boldsymbol{\Psi}_k := (\boldsymbol{I} - \hat{\boldsymbol{G}}_{\text{det},k}\hat{\boldsymbol{G}}_{\text{det},k}')$. Notice that $\boldsymbol{\Psi}_1 = \boldsymbol{I}$. Compute

$$\hat{\boldsymbol{D}}_k = \boldsymbol{\Psi}_k \left( \frac{1}{\alpha} \sum_{t=(k-1)\alpha+1}^{k\alpha} \boldsymbol{y}_t \boldsymbol{y}_t' \right) \boldsymbol{\Psi}_k$$

2. Find the $k$-th cluster, $\hat{\mathcal{G}}_k$: let $\hat{\lambda}_i = \lambda_i(\hat{\boldsymbol{D}}_k)$;
   (a) find the index $\hat{r}_k$ for which $\frac{\hat{\lambda}_1}{\hat{\lambda}_{\hat{r}_k}} \leq \hat{g}$ and either $\frac{\hat{\lambda}_1}{\hat{\lambda}_{\hat{r}_k+1}} > \hat{g}$ or $\hat{\lambda}_{\hat{r}_k+1} < \lambda_{\text{thresh}}$;
   (b) set $\hat{\mathcal{G}}_k = \{\hat{r}_* + 1, \hat{r}_* + 2, \ldots, \hat{r}_* + \hat{r}_k\}$ where $\hat{r}_* := \sum_{j=1}^{k-1} \hat{r}_j$;
   (c) if $\hat{\lambda}_{\hat{r}_k+1} < \lambda_{\text{thresh}}$, update the flag $\text{Stop} \leftarrow 1$

3. Compute $\hat{\boldsymbol{G}}_k \leftarrow \text{eigenvectors}(\hat{\boldsymbol{D}}_k, \hat{r}_k)$; increment $k$

Until $\text{Stop} == 1$.
Set $\hat{\vartheta} \leftarrow k$. Output $\hat{\boldsymbol{P}} \leftarrow [\hat{\boldsymbol{G}}_1 \cdots \hat{\boldsymbol{G}}_{\hat{\vartheta}}]$.
$\text{eigenvectors}(\boldsymbol{\mathcal{M}}, r)$ returns a basis matrix for the span of the top $r$ eigenvectors of $\boldsymbol{\mathcal{M}}$.

---

### 3.2 Cluster-EVD algorithm

The cluster-EVD approach is summarized in Algorithm 1. I Its main idea is as follows. We start by computing the empirical covariance matrix of the first set of $\alpha$ observed data points, $\hat{\boldsymbol{D}}_1 := \frac{1}{\alpha} \sum_{t=1}^{\alpha} \boldsymbol{y}_t \boldsymbol{y}_t'$. Let $\hat{\lambda}_i$ denote its $i$-th largest eigenvalue. To estimate the first cluster, $\hat{\mathcal{G}}_1$, we start with the index of the first (largest) eigenvalue and keep adding indices of the smaller eigenvalues to it until the ratio of the maximum to the minimum eigenvalue exceeds $\hat{g}$ or until the minimum eigenvalue goes below a "zero threshold", $\lambda_{\text{thresh}}$. Then, we estimate the first cluster's subspace, $\text{range}(\boldsymbol{G}_1)$ by computing the top $\hat{r}_1$ eigenvectors of $\hat{\boldsymbol{D}}_1$. To get the second cluster and its subspace, we project the next set of $\alpha$ $\boldsymbol{y}_t$'s orthogonal to $\hat{\boldsymbol{G}}_1$ followed by repeating the above procedure. This is repeated for each $k > 1$. The algorithm stops when $\hat{\lambda}_{\hat{r}_k+1} < \lambda_{\text{thresh}}$.

Algorithm 1 is related to, but significantly different from, the ones introduced in [3, 5] for the subspace deletion step of ReProCS. The one introduced in [3] assumed that the clusters were known to the algorithm (which is unrealistic). The one studied in [5] has an automatic cluster estimation approach, but, one that needs a larger lower bound on $\alpha$ compared to what Algorithm 1 needs.

### 3.3 Main result

We give the performance guarantee for Algorithm 1 here. Its parameters are set as follows. We set $\hat{g}$ to a value that is a little larger than $g$. This is needed to allow for the fact that $\hat{\lambda}_i$ is not equal to the $i$-th eigenvalue of $\boldsymbol{\Lambda}$ but is within a small margin of it. For the same reason, we need to also use a nonzero "zeroing" threshold, $\lambda_{\text{thresh}}$, that is larger than zero but smaller than $\lambda^-$. We set $\alpha$ large enough to ensure that $\text{SE}(\hat{\boldsymbol{P}}, \boldsymbol{P}) \leq r\zeta$ holds with a high enough probability.

**Theorem 3.3** (cluster-EVD result). *Consider Algorithm 1. Pick a $\zeta$ so that $r^2\zeta \leq 0.0001$, and $r^2\zeta f \leq 0.01$. Suppose that $\boldsymbol{y}_t$'s satisfy (2) and the following hold.*

1. *Assumption 1.1 and Assumption 3.2 on $\boldsymbol{\ell}_t$ hold with $\chi^+$ satisfying $\chi^+ \leq \min(1 - r\zeta - \frac{0.08}{0.25}, \frac{g^+ - 0.0001}{1.01g^+ + 0.0001} - 0.0001)$. Define*

$$\alpha_0 := C\eta^2 \frac{r^2(11\log n + \log \vartheta)}{(r\zeta)^2} \max(g^+, qg^+,$$

$$q^2 f, q(r\zeta)f, (r\zeta)^2 f, q\sqrt{fg^+}, (r\zeta)\sqrt{fg^+})^2, \ C := \frac{32 \cdot 16}{0.01^2}.$$

2. *Assumption 1.2 on $\boldsymbol{M}_t$ holds with $\alpha \geq \alpha_0$ and with $\beta$ satisfying*

$$\frac{\beta}{\alpha} \leq \left( \frac{1 - r\zeta - \chi^+}{2} \right)^2 \min\left( \frac{(r_k\zeta)^2}{4.1(qg^+)^2}, \frac{(r_k\zeta)}{q^2 f} \right).$$

*3. Set algorithm parameters $\hat{g} = 1.01g^+ + 0.0001$, $\lambda_{\text{thresh}} = 0.95\lambda^-$ and $\alpha \geq \alpha_0$.*

*Then, with probability at least $1 - 12n^{-10}$, $\text{SE}(\hat{\boldsymbol{P}}, \boldsymbol{P}) \leq r\zeta$.*

*Proof:* The proof is given in Section 9 in Supplementary Material.

We can also get corollaries for PCA-missing and PCA-SDDC for cluster-EVD. We have given one specific value for $\hat{g}$ and $\lambda_{\text{thresh}}$ in Theorem 3.3 for simplicity. One can, in fact, set $\hat{g}$ to be anything that satisfies (12) given in Supplementary Material and one can set $\lambda_{\text{thresh}}$ to be anything satisfying $5r\zeta\lambda^- \leq \lambda_{\text{thresh}} \leq 0.95\lambda^-$. Also, it should be possible to reduce the sample complexity of cluster-EVD to $c \max(q^2(g^+)^2 r \log n, (g^+)^2(r + \log n))$ using the approach explained in Sec. 2.

## 4 Discussion

**Comparing simple-EVD and cluster-EVD.** Consider the lower bounds on $\alpha$. In the cluster-EVD (c-EVD) result, Theorem 3.3, if $q$ is small enough (e.g., if $q \leq 1/\sqrt{f}$), and if $(r^2\zeta)f \leq 0.01$, it is clear that the maximum in the $\max(.,.,.,.)$ expression is achieved by $(g^+)^2$. Thus, in this regime, c-EVD needs $\alpha \geq C \frac{r^2(11 \log n + \log \vartheta)}{(r\zeta)^2} g^2$ and its sample complexity is $\vartheta\alpha$. In the EVD result (Theorem 2.1), $g^+$ gets replaced by $f$ and $\vartheta$ by 1, and so, its sample complexity, $\alpha \geq C \frac{r^2 11 \log n}{(r\zeta)^2} f^2$. In situations where the condition number $f$ is very large but $g^+$ is much smaller and $\vartheta$ is small (the clustering assumption holds well), the sample complexity of c-EVD will be much smaller than that of simple-EVD. However, notice that, the lower bound on $\alpha$ for simple-EVD holds for any $q < 1$ and for any $\zeta$ with $r\zeta < 0.01$ while the c-EVD lower bound given above holds only when $q$ is small enough, e.g., $q = O(1/\sqrt{f})$, and $\zeta$ is small enough, e.g., $r\zeta = O(1/f)$. This tighter bound on $\zeta$ is needed because the error of the $k$-th step of c-EVD depends on the errors of the previous steps times $f$. Secondly, the c-EVD result also needs $\chi^+$ and $\vartheta$ to be small (clustering assumption holds well), whereas, for simple-EVD, by definition, $\chi^+ = 0$ and $\vartheta = 1$. Another thing to note is that the constants in both lower bounds are very large with the c-EVD one being even larger.

To compare the upper bounds on $\beta$, assume that the same $\alpha$ is used by both, i.e., $\alpha = \max(\alpha_0(\text{EVD}), \alpha_0(\text{c-EVD}))$. As long as $r_k$ is large enough, $\chi^+$ is small enough, and $g$ is small enough, the upper bound on $\beta$ needed by the c-EVD result is significantly looser. For example, if $\chi^+ = 0.2$, $\vartheta = 2$, $r_k = r/2$, then c-EVD needs $\beta \leq (0.5 \cdot 0.79 \cdot 0.5)^2 \frac{(r\zeta)^2}{4.1q^2 g^2} \alpha$ while simple-EVD needs $\beta \leq (0.5 \cdot 0.99)^2 \frac{(r\zeta)^2}{4.1q^2 f^2} \alpha$. If $g = 3$ but $f = 100$, clearly the c-EVD bound is looser.

**Comparison with other results for PCA-SDDC and PCA-missing.** To our knowledge, there is no other result for correlated-PCA. Hence, we provide comparisons of the corollaries given above for the PCA-missing and PCA-SDDC special cases with works that also study these or related problems. An alternative solution for either PCA-missing or PCA-SDDC is to first recover the entire matrix $\boldsymbol{L}$ and then compute its subspace via SVD on the estimated $\boldsymbol{L}$. For the PCA-missing problem, this can be done by using any of the low-rank matrix completion techniques, e.g., nuclear norm minimization (NNM) [13] or alternating minimization (Alt-Min-MC) [23]. Similarly, for PCA-SDDC, this can be done by solving any of the recent provably correct RPCA techniques such as principal components' pursuit (PCP) [14, 15, 16] or alternating minimization (Alt-Min-RPCA) [17].

However, as explained earlier doing the above has two main disadvantages. The first is that it is much slower (see Sec. 5). The difference in speed is most dramatic when solving the matrix-sized convex programs such as NNM or PCP, but even the Alt-Min methods are slower. If we use the time complexity from [17], then finding the span of the top $k$ singular vectors of an $n \times m$ matrix takes $O(nmk)$ time. Thus, if $\vartheta$ is a constant, both simple-EVD and c-EVD need $O(n\alpha r)$ time, whereas, Alt-Min-RPCA needs $O(n\alpha r^2)$ time per iteration [17]. The second disadvantage is that the above methods for MC or RPCA need more assumptions to provably correctly recover $\boldsymbol{L}$. All the above methods need an incoherence assumption on both the left singular vectors, $\boldsymbol{P}$, and the right singular vectors, $\boldsymbol{V}$, of $\boldsymbol{L}$. Of course, it is possible that, if one studies these methods with the goal of only recovering the column space of $\boldsymbol{L}$ correctly, the incoherence assumption on the right singular vectors is not needed. From simulation experiments (see Sec. 5), the incoherence of the left singular vectors is definitely needed. On the other hand, for the PCA-SDDC problem, simple-EVD or c-EVD do not even need the incoherence assumption on $\boldsymbol{P}$.

The disadvantage of both EVD and c-EVD, or in fact of any solution for the PCA problem, is that they work only when $q$ is small enough (the corrupting noise is small compared to $\boldsymbol{\ell}_t$).

|  | Mean Subspace Error (SE) | | | | Average Execution Time | | | |
|---|---|---|---|---|---|---|---|---|
|  | c-EVD | EVD | PCP | A-M-RPCA | c-EVD | EVD | PCP | A-M-RPCA |
| Expt 1 | 0.0908 | 0.0911 | 1.0000 | 1.0000 | 0.0549 | 0.0255 | 0.2361 | 0.0810 |
| Expt 2 | 0.3626 | 0.3821 | 0.4970 | 0.4846 | 0.0613 | 0.0223 | 1.6784 | 5.5144 |

Table 1: Comparison of $\text{SE}(\hat{\boldsymbol{P}}, \boldsymbol{P})$ and execution time (in seconds). A-M-RPCA: Alt-Min-RPCA. Expt 1: simulated data, Expt 2: lake video with simulated foreground.

## 5 Numerical Experiments

We use the PCA-SDDC problem as our case study example. We compare EVD and cluster-EVD (c-EVD) with PCP [15], solved using [24], and with Alt-Min-RPCA [17] (implemented using code from the authors' webpage). For both PCP and Alt-Min-RPCA, $\hat{\boldsymbol{P}}$ is recovered as the top $r$ eigenvectors of of the estimated $\boldsymbol{L}$. To show the advantage of EVD or c-EVD, we let $\boldsymbol{\ell}_t = \boldsymbol{P}\boldsymbol{a}_t$ with columns of $\boldsymbol{P}$ being sparse. These were chosen as the first $r = 5$ columns of the identity matrix. We generate $\boldsymbol{a}_t$'s iid uniformly with zero mean and covariance matrix $\boldsymbol{\Lambda} = diag(100, 100, 100, 0.1, 0.1)$. Thus the condition number $f = 1000$. The clustering assumption holds with $\vartheta = 2$, $g^+ = 1$ and $\chi^+ = 0.001$. The noise $\boldsymbol{w}_t$ is generated as $\boldsymbol{w}_t = \boldsymbol{I}_{\mathcal{T}_t} \boldsymbol{M}_{s,t} \boldsymbol{\ell}_t$ with $\mathcal{T}_t$ generated to satisfy Assumption 1.3 with $s = 5$, $\rho = 2$, and $\tilde{\beta} = 1$; and the entries of $\boldsymbol{M}_{s,t}$ being iid $\mathcal{N}(0, q^2)$ with $q = 0.01$. We used $n = 500$. EVD and c-EVD (Algorithm 1) were implemented with $\alpha = 300$, $\lambda_{\text{thresh}} = 0.095$, $\hat{g} = 3$. 10000-time Monte Carlo averaged values of $\text{SE}(\hat{\boldsymbol{P}}, \boldsymbol{P})$ and execution time are shown in the first row of Table 1. Since the columns of $\boldsymbol{P}$ are sparse, both PCP and Alt-Min-RPCA fail. Both have average SE close to one whereas the average SE of c-EVD and EVD is 0.0908 and 0.0911 respectively. Also, both EVD and c-EVD are much faster than the other two. We also did an experiment with the settings of this experiment, but with $\boldsymbol{P}$ dense. In this case, EVD and c-EVD errors were similar, but PCP and Alt-Min-RPCA errors were less than $10^{-5}$.

For our second experiment, we used images of a low-rankified real video sequence as $\boldsymbol{\ell}_t$'s. We chose the escalator sequence from `http://perception.i2r.a-star.edu.sg/bk_model/bk_index.html` since the video changes are only in the region where the escalator moves (and hence can be modeled as being sparse). We made it exactly low-rank by retaining its top 5 eigenvectors and projecting onto their subspace. This resulted in a data matrix $\boldsymbol{L}$ of size $n \times 2\alpha$ with $n = 20800$ and $\alpha = 50$, of rank $r = 5$. We overlaid a simulated moving foreground block on it. The intensity of the moving block was controlled to ensure that $q$ is small. We used $\alpha = 50$ for EVD and c-EVD. We let $\boldsymbol{P}$ be the eigenvectors of the low-rankified video with nonzero eigenvalues and computed $\text{SE}(\hat{\boldsymbol{P}}, \boldsymbol{P})$. The errors and execution time are displayed in the second row of Table 1. Since $n$ is very large, the difference in speed is most apparent in this case.

Thus c-EVD outperforms PCP and AltMinRPCA when columns of $\boldsymbol{P}$ are sparse. It also outperforms EVD but the advantage in mean error is not as much as our theorems predict. One reason is that the constant in the required lower bounds on $\alpha$ is very large. It is hard to pick an $\alpha$ that is this large and still only $O(\log n)$ unless $n$ is very large. Secondly, both guarantees are only sufficient conditions.

## 6 Conclusions and Future Work

We studied the problem of PCA in noise that is correlated with the data (data-dependent noise). We obtained sample complexity bounds for the most commonly used PCA solution, simple EVD. We also developed and analyzed a generalization of EVD, called cluster-EVD, that has lower sample complexity under extra assumptions. We provided a detailed comparison of our results with those for other approaches to solving its example applications - PCA with missing data and PCA with sparse data-dependent corruptions.

We used matrix Hoeffding [20] to obtain our results. As explained in Sec. 2, we can significantly improve the sample complexity bounds if this is replaced by [21, Theorem 5.39] and matrix Bernstein at appropriate places. We have obtained such a result in ongoing work [25]. Moreover, as done in [5] (for ReProCS), the mutual independence of $\boldsymbol{\ell}_t$'s can be easily replaced by a more practical assumption of $\boldsymbol{\ell}_t$'s following autoregressive model with almost no change to our assumptions. Thirdly, by generalizing the proof techniques developed here, we can also study the problem of correlated-PCA with partial subspace knowledge. The solution to the latter problem helps to greatly simplify the proof of correctness of ReProCS for online dynamic RPCA. The boundedness assumption on $\boldsymbol{\ell}_t$'s can be replaced by a Gaussian or a well-behaved sub-Gaussian assumption but this will increase the sample complexity to $O(n)$. Finally, an open-ended question is how we relax Assumption 1.2 on $\boldsymbol{M}_t$ and still get results similar to Theorem 2.1 or Theorem 3.3.

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
