[Supplementary Material]

# Supplementary Material

## 7 More examples of Assumption 1.2

Assumption 1.3 is one simple example of a support change model that ensures that, if $M_{2,t} = I_{\mathcal{T}_t}$, the assumption on $M_{2,t}$ given in Assumption 1.2 holds. If instead of one object, there are $k$ objects, and each of their supports satisfies Assumption 1.3, then again, with some modifications, it is possible to show that both the PCA-missing and PCA-SDDC problems satisfy Assumption 1.2. Moreover, notice that Assumption 1.3 does not require the entries in $\mathcal{T}_t$ to be contiguous at all (they need not correspond to the support of one or a few objects). Similarly, we can replace the condition that $\mathcal{T}_t$ be constant for at most $\tilde{\beta}$ time instants in Assumption 1.3 by $|\{t : \mathcal{T}_t = \mathcal{T}^{[k]}\}| \leq \tilde{\beta}$.

Thirdly, the requirement of the object(s) always moving in one direction may seem too stringent. As explained in [4, Lemma 9.4], a Bernoulli-Gaussian "constant velocity with random acceleration" motion model will also work whp. It allows the object to move at each frame with probability $p$ and not move with probability $1 - p$ independent of past or future frames; when the object moves, it moves with an iid Gaussian velocity that has mean $1.1s/\rho$ and variance $\sigma^2$; $\sigma^2$ needs to be upper bounded and $p$ needs to be lower bounded.

Lastly, if $s < c_1 \alpha$ for $c_1 \ll 1$, another model that works is that of an object of length $s$ or less moving by at least one pixel and at most $b$ pixels at each time [4, Lemma 9.5].

## 8 Proof of Theorem 2.1

This result also follows as a corollary of Theorem 3.3. We prove it separately first since its proof is short and and less notation-ally intensive. It will help understand the proof of Theorem 3.3 much more easily. Both results rely on the $\sin\theta$ theorem reviewed next.

### 8.1 $\sin\theta$ theorem

Davis and Kahan's $\sin\theta$ theorem [19] studies the rotation of eigenvectors by perturbation.

**Theorem 8.1** ($\sin\theta$ theorem [19]). *Consider two Hermitian matrices $D$ and $\hat{D}$. Suppose that $D$ can be decomposed as*

$$D = [\ E \quad E_\perp\ ] \begin{bmatrix} A & 0 \\ 0 & A_\perp \end{bmatrix} \begin{bmatrix} E' \\ E_\perp' \end{bmatrix}$$

*where $[E\ E_\perp]$ is an orthonormal matrix. Suppose that $\hat{D}$ can be decomposed as*

$$\hat{D} = [\ F \quad F_\perp\ ] \begin{bmatrix} \Lambda & 0 \\ 0 & \Lambda_\perp \end{bmatrix} \begin{bmatrix} F' \\ F_\perp' \end{bmatrix}$$

*where $[F\ F_\perp]$ is another orthonormal matrix and is such that $\mathrm{rank}(F) = \mathrm{rank}(E)$. Let $H := \hat{D} - D$ denote the perturbation. If $\lambda_{\min}(A) > \lambda_{\max}(\Lambda_\perp)$, then*

$$\|(I - FF')E\| \leq \frac{\|H\|}{\lambda_{\min}(A) - \lambda_{\max}(\Lambda_\perp)}.$$

Let $r = \mathrm{rank}(E)$. Suppose that $F$ is the matrix of top $r$ eigenvectors of $\hat{D}$. Then $\Lambda$ and $\Lambda_\perp$ are diagonal and $\lambda_{\max}(\Lambda_\perp) = \lambda_{r+1}(\hat{D}) \leq \lambda_{r+1}(D) + \|H\|$. The inequality follows using Weyl's inequality. Suppose also that $\lambda_{\min}(A) > \lambda_{\max}(A_\perp)$. Then, (i) $\lambda_r(D) = \lambda_{\min}(A)$ and $\lambda_{r+1}(D) = \lambda_{\max}(A_\perp)$ and (ii) $\mathrm{range}(E)$ is equal to the span of the top $r$ eigenvectors of $D$. Thus, $\lambda_{\max}(\Lambda_\perp) \leq \lambda_{\max}(A_\perp) + \|H\|$. With this we have the following corollary.

**Corollary 8.2.** *Consider a Hermitian matrix $D$ and its perturbed version $\hat{D}$. Suppose that $D$ can be decomposed as*

$$D = [\ E \quad E_\perp\ ] \begin{bmatrix} A & 0 \\ 0 & A_\perp \end{bmatrix} \begin{bmatrix} E' \\ E_\perp' \end{bmatrix}$$

*where $E$ is a basis matrix. Let $F$ denote the matrix containing the top $\mathrm{rank}(E)$ eigenvectors of $\hat{D}$. Let $H := \hat{D} - D$ denote the perturbation. If $\lambda_{\min}(A) - \lambda_{\max}(A_\perp) - \|H\| > 0$, then*

$$\|(I - FF')E\| \leq \frac{\|H\|}{\lambda_{\min}(A) - \lambda_{\max}(A_\perp) - \|H\|}.$$

*and $\mathrm{range}(E)$ is equal to the span of the top $\mathrm{rank}(E)$ eigenvectors of $D$.*

## 8.2 Proof of Theorem 2.1

We use the $\sin\theta$ theorem [19] from Corollary 8.2. Apply it with $\hat{D} = \frac{1}{\alpha}\sum_t y_t y_t'$ and $D = \frac{1}{\alpha}\sum_t \ell_t \ell_t'$. Thus, $F = \hat{P}$. Recall that $a_t = P'\ell_t$. Then, $D$ can be decomposed as $P(\frac{1}{\alpha}\sum_t a_t a_t')P' + P_\perp 0 P_\perp'$, and so we have $E = P$, $A = \frac{1}{\alpha}\sum_t a_t a_t'$ and $A_\perp = 0$. Moreover, it is easy to see that the perturbation $H := \frac{1}{\alpha}\sum_t y_t y_t' - \frac{1}{\alpha}\sum_t \ell_t \ell_t'$ satisfies

$$H = \frac{1}{\alpha}\sum_t \ell_t w_t' + \frac{1}{\alpha}\sum_t w_t \ell_t' + \frac{1}{\alpha}\sum_t w_t w_t'. \tag{7}$$

Thus,

$$\mathrm{SE}(\hat{P}, P)$$
$$\leq \frac{2\|\frac{1}{\alpha}\sum_t \ell_t w_t'\| + \|\frac{1}{\alpha}\sum_t w_t w_t'\|}{\lambda_r(\frac{1}{\alpha}\sum_t \ell_t \ell_t') - (2\|\frac{1}{\alpha}\sum_t \ell_t w_t'\| + \|\frac{1}{\alpha}\sum_t w_t w_t'\|)}$$

if the denominator is positive.

**Remark 8.3.** *Because $w_t$ is correlated with $\ell_t$, the $\ell_t w_t'$ terms are the dominant ones in the perturbation expression given in (7). If they were uncorrelated, these two terms would be close to zero whp due to law of large numbers and the $w_t w_t'$ term would be the dominant one.*

In the next lemma, we bound the terms in the bound on $\mathrm{SE}(\hat{P}, P)$ using the matrix Hoeffding inequality [20].

**Lemma 8.4.** *Let $\epsilon = 0.01 r\zeta\lambda^-$.*

1. *With probability at least $1 - 2n\exp\left(-\alpha\frac{\epsilon^2}{32(\eta r q\lambda^+)^2}\right)$,*

$$\|\frac{1}{\alpha}\sum_t \ell_t w_t'\| \leq q\lambda^+\sqrt{\frac{\beta}{\alpha}} + \epsilon = [qf\sqrt{\frac{\beta}{\alpha}} + 0.01 r\zeta]\lambda^-$$

2. *With probability at least $1 - 2n\exp(-\frac{\alpha\epsilon^2}{32(\eta r q^2\lambda^+)^2})$,*

$$\|\frac{1}{\alpha}\sum_t w_t w_t'\| \leq \frac{\beta}{\alpha}q^2\lambda^+ + \epsilon = [q^2 f\frac{\beta}{\alpha} + 0.01 r\zeta]\lambda^-$$

3. *With probability at least $1 - 2n\exp(-\frac{\alpha\epsilon^2}{32(\eta r\lambda^+)^2})$,*

$$\lambda_r(\frac{1}{\alpha}\sum_t \ell_t \ell_t') \geq (1 - (r\zeta)^2)\lambda^- - \epsilon$$

*Proof.* This follows by using Lemma 9.6 given later with $G_{\mathrm{cur}} \equiv P$, $G_{\mathrm{det}} \equiv [.]$, $G_{\mathrm{undet}} \equiv [.]$, $\zeta_{\mathrm{det}} \equiv 0$, $r\zeta \equiv 0$, $r_{\mathrm{cur}} = r$, $g \equiv f$, $\chi \equiv 0$, $\vartheta \equiv 1$. ⊠

Using this lemma to bound the subspace error terms, followed by using the bounds on $\beta/\alpha$ and $\zeta$, we conclude the following: w.p. at least $1 - 2n\exp\left(-\alpha\frac{\epsilon^2}{32(\eta r q\lambda^+)^2}\right) - 2n\exp(-\frac{\alpha\epsilon^2}{32(\eta r q^2\lambda^+)^2}) - 2n\exp(-\frac{\alpha\epsilon^2}{32(\eta r\lambda^+)^2})$,

$$\mathrm{SE}(\hat{P}, P)$$
$$\leq \frac{2qf\sqrt{\frac{\beta}{\alpha}} + q^2 f\frac{\beta}{\alpha} + 0.03 r\zeta}{1 - (r\zeta)^2 - 0.01 r\zeta - (2qf\sqrt{\frac{\beta}{\alpha}} + q^2 f\frac{\beta}{\alpha} + 0.03 r\zeta)}$$
$$\leq \frac{0.75(1 - r\zeta)r\zeta + 0.03 r\zeta}{1 - r\zeta} < r\zeta$$

Using the bound $\alpha \geq \alpha_0$ from the theorem, the probability of the above event is at least $1 - 6n^{-10}$. We get this by bounding each of the three negative terms in the probability expression by $-2n^{-10}$. We work this out for the first term: $\alpha\frac{\epsilon^2}{32(\eta r q\lambda^+)^2} \geq \frac{32\cdot 11}{(0.01)^2}\frac{\eta^2 r^2(\log n)}{(r\zeta)^2}(qf)^2\frac{(0.01 r\zeta\lambda^-)^2}{32\eta^2 r^2 q^2\lambda^{+2}} = 11\log n$. Thus, $2n\exp\left(-\alpha\frac{\epsilon^2}{32(\eta r q\lambda^+)^2}\right) \leq 2n\exp(-11\log n) \leq 2n^{-10}$.

# 9 Proof of Theorem 3.3

We explain the overall idea of the proof next. In Sec. 9.2, we give a sequence of lemmas in generalized form (so that they can apply to various other problems). The proof of Theorem 3.3 is given in Sec. 9.3 and follows easily by applying these. One of the lemmas of Sec. 9.2 is proved in Sec. 10 while the others are proved there itself.

## 9.1 Overall idea

We need to bound $\mathrm{SE}(\hat{P}, P)$. From Algorithm 1, $\hat{P} = [\hat{G}_1, \hat{G}_2, \dots, \hat{G}_\vartheta]$ where $\hat{G}_k$ is the matrix of top $\hat{r}_k$ eigenvectors of $\hat{D}_k$ defined in Algorithm 1. Also, $P = [G_1, G_2, \dots, G_\vartheta]$ where $G_k$ is a basis matrix with $r_k$ columns.

**Definition 9.1.** *Define* $\zeta_k := \mathrm{SE}([\hat{G}_1, \hat{G}_2, \dots, \hat{G}_k], G_k)$ *and* $\zeta_0 = 0$. *Define* $\zeta_k^+ := r_k \zeta$. *Let* $r_0 = 0$.

It is easy to see that

$$
\begin{aligned}
\mathrm{SE}(\hat{P}, P) &\le \sum_{k=1}^{\vartheta} \mathrm{SE}(\hat{P}, G_k) \\
&\le \sum_{k=1}^{\vartheta} \mathrm{SE}([\hat{G}_1, \hat{G}_2, \dots, \hat{G}_k], G_k) = \sum_{k=1}^{\vartheta} \zeta_k
\end{aligned}
\tag{8}
$$

The first inequality is triangle inequality, the second follows because $[\hat{G}_1, \hat{G}_2, \dots, \hat{G}_k]$ is orthogonal to $[\hat{G}_{k+1}, \dots G_\vartheta]$. Since $r = \sum_k r_k$, if we can show that $\zeta_k \le \zeta_k^+ = r_k \zeta$ for all $k$ we will be done.

We bound $\zeta_k$ using induction. The base case is easy and follows just from the definition, $\zeta_0 = \mathrm{SE}([.], [.]) = 0 = r_0 \zeta$. For bounding $\zeta_k$, assume that for all $i = 1, 2, \dots, k-1$, $\zeta_i \le r_i \zeta$. This implies that

$$
\begin{aligned}
&\mathrm{SE}([\hat{G}_1, \hat{G}_2, \dots, \hat{G}_{k-1}], [G_1, G_2, \dots, G_{k-1}]) \\
&\le \sum_{i=1}^{k-1} \mathrm{SE}([\hat{G}_1, \hat{G}_2, \dots, \hat{G}_{k-1}], G_i) \\
&\le \sum_{i=1}^{k-1} \zeta_i \le \sum_{i=1}^{k-1} r_i \zeta \le r \zeta
\end{aligned}
\tag{9}
$$

Using this, we will first show that $\hat{r}_k = r_k$, and then we will use this and the $\sin \theta$ result to bound $\zeta_k$.

Before proceeding further, we simplify notation.

**Definition 9.2.**

   1. *Let*

$$
\begin{aligned}
G_{\mathrm{det}} &:= [G_1, G_2, \dots, G_{k-1}], \ G_{\mathrm{cur}} := G_k, \\
G_{\mathrm{undet}} &:= [\hat{G}_{k+1}, \dots G_\vartheta]
\end{aligned}
$$

   2. *Similarly, let* $\hat{G}_{\mathrm{det}} := [\hat{G}_1, \hat{G}_2, \dots, \hat{G}_{k-1}]$, $\hat{G}_{\mathrm{cur}} := \hat{G}_k$.

   3. *Let* $\mathcal{G}_{\mathrm{det}} := \mathcal{G}_1 \cup \mathcal{G}_2 \cdots \cup \mathcal{G}_{k-1}$ *and* $\mathcal{G}_{\mathrm{cur}} = \mathcal{G}_k$.

   4. *Let* $r_{\mathrm{cur}} := r_k = \mathrm{rank}(G_k)$ *and* $\hat{r}_{\mathrm{cur}} := \hat{r}_k$.

   5. *Let* $\lambda_{\mathrm{cur}}^+ := \lambda_k^+$, $\lambda_{\mathrm{cur}}^- := \lambda_k^-$, $\lambda_{\mathrm{undet}}^+ := \lambda_{k+1}^+$

   6. *Let* $t_* = k\alpha$.

## 9.2 Main lemmas - generalized form

In this section, we give a sequence of lemmas that apply to a generic problem where $y_t = \ell_t + w_t = \ell_t + M_t \ell_t$ with $\ell_t$ satisfying Assumption 1.1; $M_t$ satisfying Assumption 1.2; and with $P$ split into three parts as $P = [G_{\mathrm{det}}, G_{\mathrm{cur}}, G_{\mathrm{undet}}]$. We can correspondingly split $\Lambda$ as $\Lambda = \mathrm{diag}(\Lambda_{\mathrm{det}}, \Lambda_{\mathrm{cur}}, \Lambda_{\mathrm{undet}})$.

We are given $\hat{\boldsymbol{G}}_{\mathrm{det}}$ that was computed using (some or all) $\boldsymbol{y}_t$'s for $t \le t_*$ and that satisfies $\zeta_{\mathrm{det}} \le r\zeta$. The goal is to estimate $\mathrm{range}(\boldsymbol{G}_{\mathrm{cur}})$ and bound the estimation error. This is done by first estimating $\hat{r}_{\mathrm{cur}}$ and then computing $\hat{\boldsymbol{G}}_{\mathrm{cur}}$ as the top $\hat{r}_{\mathrm{cur}}$ eigenvectors of

$$\hat{\boldsymbol{D}} := \frac{1}{\alpha} \sum_{t=t_*+1}^{t_*+\alpha} \boldsymbol{\Psi} \boldsymbol{y}_t \boldsymbol{y}_t' \boldsymbol{\Psi}. \tag{10}$$

To bound the estimation error, we first show that, whp, $\hat{r}_{\mathrm{cur}} = r_{\mathrm{cur}}$ and so $\hat{\mathcal{G}}_{\mathrm{cur}} = \mathcal{G}_{\mathrm{cur}}$; and then we use this to show that $\zeta_{\mathrm{cur}} \le r_{\mathrm{cur}}\zeta$.

**Definition 9.3.**

1. *Define* $\boldsymbol{\Psi} := \boldsymbol{I} - \hat{\boldsymbol{G}}_{\mathrm{det}} \hat{\boldsymbol{G}}_{\mathrm{det}}'$.

2. *Define* $\zeta_{\mathrm{det}} := \mathrm{SE}(\hat{\boldsymbol{G}}_{\mathrm{det}}, \boldsymbol{G}_{\mathrm{det}}) = \|\boldsymbol{\Psi} \boldsymbol{G}_{\mathrm{det}}\|$ *and* $\zeta_{\mathrm{det}}^+ = r\zeta$

3. *Define* $\zeta_{\mathrm{cur}} := \mathrm{SE}([\hat{\boldsymbol{G}}_{\mathrm{det}}, \hat{\boldsymbol{G}}_{\mathrm{cur}}], \boldsymbol{G}_{\mathrm{cur}})$.

4. *Let* $(\boldsymbol{\Psi} \boldsymbol{G}_{\mathrm{cur}}) \overset{\mathrm{QR}}{=} \boldsymbol{E}_{\mathrm{cur}} \boldsymbol{R}_{\mathrm{cur}}$ *denote its reduced QR decomposition. Thus $\boldsymbol{E}_{\mathrm{cur}}$ is a basis matrix whose span equals that of $(\boldsymbol{\Psi} \boldsymbol{G}_{\mathrm{cur}})$ and $\boldsymbol{R}_{\mathrm{cur}}$ is a square upper triangular matrix with $\|\boldsymbol{R}_{\mathrm{cur}}\| = \|\boldsymbol{\Psi} \boldsymbol{G}_{\mathrm{cur}}\| \le 1$.*

5. *Let* $\lambda_{\mathrm{cur}}^+ = \lambda_{\max}(\boldsymbol{\Lambda}_{\mathrm{cur}})$, $\lambda_{\mathrm{cur}}^- = \lambda_{\min}(\boldsymbol{\Lambda}_{\mathrm{cur}})$, $\lambda_{\mathrm{undet}}^+ = \lambda_{\max}(\boldsymbol{\Lambda}_{\mathrm{undet}})$.

6. *Let* $r_{\mathrm{cur}} = \mathrm{rank}(\boldsymbol{G}_{\mathrm{cur}})$. *Clearly,* $r_{\mathrm{cur}} \le r$.

**Remark 9.4.** *In special cases, $\boldsymbol{G}_{\mathrm{det}}$ (and hence $\hat{\boldsymbol{G}}_{\mathrm{det}}$) could be empty; and/or $\boldsymbol{G}_{\mathrm{undet}}$ could be empty.*

- Since $\boldsymbol{\Lambda}$ contains eigenvalues in decreasing order, when $\boldsymbol{G}_{\mathrm{undet}}$ is not empty, $\lambda^- \le \lambda_{\mathrm{undet}}^+ \le \lambda_{\mathrm{cur}}^- \le \lambda_{\mathrm{cur}}^+ \le \lambda^+$.
- When $\boldsymbol{G}_{\mathrm{undet}}$ is empty, $\lambda_{\mathrm{undet}}^+ = 0$ and $\lambda^- \le \lambda_{\mathrm{cur}}^- \le \lambda_{\mathrm{cur}}^+ \le \lambda^+$.

Using $\|\boldsymbol{R}_{\mathrm{cur}}\| = \|\boldsymbol{\Psi} \boldsymbol{G}_{\mathrm{cur}}\| \le 1$,

$$\zeta_{\mathrm{cur}} = \|(\boldsymbol{I} - \hat{\boldsymbol{G}}_{\mathrm{cur}} \hat{\boldsymbol{G}}_{\mathrm{cur}}') \boldsymbol{\Psi} \boldsymbol{G}_{\mathrm{cur}}\|$$
$$= \|(\boldsymbol{I} - \hat{\boldsymbol{G}}_{\mathrm{cur}} \hat{\boldsymbol{G}}_{\mathrm{cur}}') \boldsymbol{E}_{\mathrm{cur}} \boldsymbol{R}_{\mathrm{cur}}\|$$
$$\le \|(\boldsymbol{I} - \hat{\boldsymbol{G}}_{\mathrm{cur}} \hat{\boldsymbol{G}}_{\mathrm{cur}}') \boldsymbol{E}_{\mathrm{cur}}\| = \mathrm{SE}(\hat{\boldsymbol{G}}_{\mathrm{cur}}, \boldsymbol{E}_{\mathrm{cur}}).$$

Thus, to bound $\zeta_{\mathrm{cur}}$ we need to bound $\mathrm{SE}(\hat{\boldsymbol{G}}_{\mathrm{cur}}, \boldsymbol{E}_{\mathrm{cur}})$. $\hat{\boldsymbol{G}}_{\mathrm{cur}}$ is the matrix of top $\hat{r}_{\mathrm{cur}}$ eigenvectors of $\hat{\boldsymbol{D}}$. From its definition, $\boldsymbol{E}_{\mathrm{cur}}$ is a basis matrix with $r_{\mathrm{cur}}$ columns. Suppose for a moment that $\hat{r}_{\mathrm{cur}} = r_{\mathrm{cur}}$. Then, in order to bound $\mathrm{SE}(\hat{\boldsymbol{G}}_{\mathrm{cur}}, \boldsymbol{E}_{\mathrm{cur}})$, we can use the $\sin \theta$ result, Corollary 8.2. To do this, we need to define a matrix $\boldsymbol{D}$ so that, under appropriate assumptions, the span of its top $r_{\mathrm{cur}}$ eigenvectors equals $\mathrm{range}(\boldsymbol{E}_{\mathrm{cur}})$. For the simple EVD proof, we used $\frac{1}{\alpha} \sum_{t=t_*+1}^{t_*+\alpha} \boldsymbol{\Psi} \boldsymbol{\ell}_t \boldsymbol{\ell}_t' \boldsymbol{\Psi}$ as the matrix $\boldsymbol{D}$. However, this will not work now since $\boldsymbol{E}_{\mathrm{cur}}$ is not orthonormal to $\boldsymbol{\Psi} \boldsymbol{G}_{\mathrm{det}}$ or to $\boldsymbol{\Psi} \boldsymbol{G}_{\mathrm{undet}}$. But, instead we can use

$$\boldsymbol{D} = \boldsymbol{E}_{\mathrm{cur}} \boldsymbol{A} \boldsymbol{E}_{\mathrm{cur}}' + \boldsymbol{E}_{\mathrm{cur},\perp} \boldsymbol{A}_\perp \boldsymbol{E}_{\mathrm{cur},\perp}', \text{ where}$$

$$\boldsymbol{A} := \boldsymbol{E}_{\mathrm{cur}}'(\frac{1}{\alpha} \sum_{t=t_*+1}^{t_*+\alpha} \boldsymbol{\Psi} \boldsymbol{\ell}_t \boldsymbol{\ell}_t' \boldsymbol{\Psi}) \boldsymbol{E}_{\mathrm{cur}} \text{ and}$$

$$\boldsymbol{A}_\perp := \boldsymbol{E}_{\mathrm{cur},\perp}'(\frac{1}{\alpha} \sum_{t=t_*+1}^{t_*+\alpha} \boldsymbol{\Psi} \boldsymbol{\ell}_t \boldsymbol{\ell}_t' \boldsymbol{\Psi}) \boldsymbol{E}_{\mathrm{cur},\perp} \tag{11}$$

Now, by construction, $\boldsymbol{D}$ is in the desired form.

With the above choice of $\boldsymbol{D}$, $\boldsymbol{H} := \hat{\boldsymbol{D}} - \boldsymbol{D}$ satisfies[1] $\boldsymbol{H} = \mathrm{term1} + \mathrm{term1}' + \mathrm{term2} + \mathrm{term3} + \mathrm{term3}'$ where $\mathrm{term1} := \frac{1}{\alpha} \sum_t \boldsymbol{\Psi} \boldsymbol{\ell}_t \boldsymbol{w}_t' \boldsymbol{\Psi}$, $\mathrm{term2} := \frac{1}{\alpha} \sum_t \boldsymbol{\Psi} \boldsymbol{w}_t \boldsymbol{w}_t' \boldsymbol{\Psi}$ and $\mathrm{term3} = \boldsymbol{E}_{\mathrm{cur}} \boldsymbol{E}_{\mathrm{cur}}'(\frac{1}{\alpha} \sum_t \boldsymbol{\Psi} \boldsymbol{\ell}_t \boldsymbol{\ell}_t' \boldsymbol{\Psi}) \boldsymbol{E}_{\mathrm{cur},\perp} \boldsymbol{E}_{\mathrm{cur},\perp}'$.

Thus, using the above along with Corollary 8.2, we can conclude the following.

**Fact 9.5.**

1. *If* $\hat{r}_{\mathrm{cur}} = r_{\mathrm{cur}}$, *and* $\lambda_{\min}(\boldsymbol{A}) - \lambda_{\max}(\boldsymbol{A}_\perp) - \|\boldsymbol{H}\| > 0$,

$$\zeta_{\mathrm{cur}} \leq \mathrm{SE}(\hat{\boldsymbol{G}}_{\mathrm{cur}}, \boldsymbol{E}_{\mathrm{cur}}) \leq \frac{\|\boldsymbol{H}\|}{\lambda_{\min}(\boldsymbol{A}) - \lambda_{\max}(\boldsymbol{A}_\perp) - \|\boldsymbol{H}\|}.$$

2. *Let* $\boldsymbol{Q} := \boldsymbol{E}_{\mathrm{cur}}\boldsymbol{E}_{\mathrm{cur}}'(\frac{1}{\alpha}\sum_t \boldsymbol{\Psi}\boldsymbol{\ell}_t\boldsymbol{\ell}_t'\boldsymbol{\Psi})\boldsymbol{E}_{\mathrm{cur},\perp}\boldsymbol{E}_{\mathrm{cur},\perp}'$. *We have*

$$\|\boldsymbol{H}\| \leq 2\|\frac{1}{\alpha}\sum_t \boldsymbol{\Psi}\boldsymbol{\ell}_t\boldsymbol{w}_t'\| + \|\frac{1}{\alpha}\sum_t \boldsymbol{w}_t\boldsymbol{w}_t'\| + 2\|\boldsymbol{Q}\|.$$

The next lemma bounds the RHS terms in the above lemma and a few other quantities needed for showing $\hat{r}_{\mathrm{cur}} = r_{\mathrm{cur}}$.

**Lemma 9.6.** *(1) Assume that* $\boldsymbol{y}_t = \boldsymbol{\ell}_t + \boldsymbol{w}_t = \boldsymbol{\ell}_t + \boldsymbol{M}_t\boldsymbol{\ell}_t$ *with* $\boldsymbol{\ell}_t$ *satisfying Assumption 1.1 and* $\boldsymbol{M}_t$ *satisfying Assumption 1.2.*

*(2) Assume that we are given* $\hat{\boldsymbol{G}}_{\mathrm{det}}$ *that was computed using (some or all)* $\boldsymbol{y}_t$*'s for* $t \leq t_*$ *and that satisfies* $\zeta_{\mathrm{det}} \leq r\zeta$.

*Define* $g := \lambda_{\mathrm{cur}}^+/\lambda_{\mathrm{cur}}^-$, $\chi := \lambda_{\mathrm{undet}}^+/\lambda_{\mathrm{cur}}^-$. *Set* $\epsilon := 0.01 r_{\mathrm{cur}}\zeta\lambda_{\mathrm{cur}}^-$.

*Then, the following hold:*

1. *Let* $p_1 := 2n\exp(-\frac{\alpha\epsilon^2}{32b_{prob}^2})$ *where* $b_{prob} := \eta r q((r\zeta)\lambda^+ + \lambda_{\mathrm{cur}}^+ + (r\zeta)\sqrt{\lambda^+\lambda_{\mathrm{cur}}^+} + \sqrt{\lambda^+\lambda_{\mathrm{cur}}^+})$. *Conditioned on* $\{\zeta_{\mathrm{det}} \leq r\zeta\}$, *with probability at least* $1 - p_1$

$$\|\frac{1}{\alpha}\sum_t \boldsymbol{\Psi}\boldsymbol{\ell}_t\boldsymbol{w}_t'\| \leq q((r\zeta)\lambda^+ + \lambda_{\mathrm{cur}}^+)\sqrt{\frac{\beta}{\alpha}} + \epsilon$$

$$\leq [q(r\zeta)f\sqrt{\frac{\beta}{\alpha}} + qg\sqrt{\frac{\beta}{\alpha}} + 0.01 r_{\mathrm{cur}}\zeta]\lambda_{\mathrm{cur}}^-.$$

2. *Let* $p_2 := 2n\exp(-\frac{\alpha\epsilon^2}{32(q^2\eta r\lambda^+)^2})$. *Conditioned on* $\{\zeta_{\mathrm{det}} \leq r\zeta\}$, *with probability (w.p.) at least* $1 - p_2$,

$$\|\frac{1}{\alpha}\sum_t \boldsymbol{w}_t\boldsymbol{w}_t'\| \leq \frac{\beta}{\alpha}q^2\lambda^+ + \epsilon \leq [\frac{\beta}{\alpha}q^2 f + 0.01 r_{\mathrm{cur}}\zeta]\lambda_{\mathrm{cur}}^-.$$

3. *Let* $p_3 := 2n\exp(-\frac{\alpha\epsilon^2}{32b_{prob}^2})$ *with* $b_{prob} := \eta r((r\zeta)^2\lambda^+ + \lambda_{\mathrm{cur}}^+ + 2(r\zeta)\sqrt{\lambda^+\lambda_{\mathrm{cur}}^+})$. *Conditioned on* $\{\zeta_{\mathrm{det}} \leq r\zeta\}$, *with probability at least* $1 - p_3$,

$$\|\boldsymbol{E}_{\mathrm{cur}}\boldsymbol{E}_{\mathrm{cur}}'(\frac{1}{\alpha}\boldsymbol{\Psi}\boldsymbol{\ell}_t\boldsymbol{\ell}_t'\boldsymbol{\Psi})\boldsymbol{E}_{\mathrm{cur},\perp}\boldsymbol{E}_{\mathrm{cur},\perp}'\|$$

$$\leq (r\zeta)^2\lambda^+ + \frac{(r\zeta)^2}{\sqrt{1-(r\zeta)^2}}\lambda_{\mathrm{undet}}^+ + \epsilon$$

$$\leq [(r\zeta)^2 f + \frac{(r\zeta)^2}{\sqrt{1-(r\zeta)^2}}\chi + 0.01 r_{\mathrm{cur}}\zeta]\lambda_{\mathrm{cur}}^-.$$

4. *Conditioned on* $\{\zeta_{\mathrm{det}} \leq r\zeta\}$, *w.p. at least* $1 - p_3$,

$$\lambda_{\min}(\boldsymbol{A}) \geq (1-(r\zeta)^2)\lambda_{\mathrm{cur}}^- - \epsilon$$
$$= [1-(r\zeta)^2 - 0.01 r_{\mathrm{cur}}\zeta]\lambda_{\mathrm{cur}}^-$$

5. *Conditioned on* $\{\zeta_{\mathrm{det}} \leq r\zeta\}$, *w.p. at least* $1 - p_3$,

$$\lambda_{\max}(\boldsymbol{A}_\perp) \leq ((r\zeta)^2\lambda^+ + \lambda_{\mathrm{undet}}^+) + \epsilon$$
$$\leq [(r\zeta)^2 f + \chi + 0.01 r_{\mathrm{cur}}\zeta]\lambda_{\mathrm{cur}}^-.$$

6. *Conditioned on* $\{\zeta_{\text{det}} \leq r\zeta\}$, *with probability at least* $1 - p_3$,

$$\lambda_{\max}(\boldsymbol{A}_\perp) \geq (1 - (r\zeta)^2 - \frac{(r\zeta)^2}{\sqrt{1 - (r\zeta)^2}})\lambda^+_{\text{undet}} - \epsilon.$$

7. *Conditioned on* $\{\zeta_{\text{det}} \leq r\zeta\}$, *w.p. at least* $1 - p_3$,

$$\lambda_{\max}(\boldsymbol{A}) \geq (1 - (r\zeta)^2)\lambda^+_{\text{cur}} - \epsilon$$
$$= [(1 - (r\zeta)^2)g - 0.01r_{\text{cur}}\zeta]\lambda^-_{\text{cur}}.$$

8. *Conditioned on* $\{\zeta_{\text{det}} \leq r\zeta\}$, *w.p. at least* $1 - p_3$,

$$\lambda_{\max}(\boldsymbol{A}) \leq \lambda^+_{\text{cur}} + (r\zeta)^2\lambda^+ + \frac{1}{1 - r^2\zeta^2}(r\zeta)^2\lambda^+_{\text{undet}} + \epsilon$$

$$\leq [g + (r\zeta)^2 f + \frac{(r\zeta)^2}{1 - (r\zeta)^2}\chi + 0.01r_{\text{cur}}\zeta]\lambda^-_{\text{cur}}.$$

*Proof.* The proof is in Section 10. ⊠

**Corollary 9.7.** *Consider the setting of Lemma 9.6. Assume*

1. $r(r\zeta) \leq 0.0001$, *and* $r(r\zeta)f \leq 0.01$. *Since* $r_{\text{cur}} \leq r$, *this implies that* $r_{\text{cur}}\zeta \leq 0.0001$, *and*

2. $\beta \leq \left(\frac{(1 - r_{\text{cur}}\zeta - \chi)}{2}\right)^2 \min\left(\frac{(r_{\text{cur}}\zeta)^2}{4.1q^2g^2}, \frac{(r_{\text{cur}}\zeta)}{q^2 f}\right)\alpha.$

*Using these and using* $g \geq 1$, $g \leq f$, $\chi \leq 1$ *(these hold by definition), with probability at least* $1 - p_1 - p_2 - 4p_3$,

$$\|\boldsymbol{H}\| \leq [2.02qg\sqrt{\frac{\beta}{\alpha}} + \frac{\beta}{\alpha}q^2 f + 0.08r_{\text{cur}}\zeta]\lambda^-_{\text{cur}}$$

$$\leq [0.75(1 - r\zeta - \chi)r_{\text{cur}}\zeta + 0.08r_{\text{cur}}\zeta]\lambda^-_{\text{cur}}$$

$$\leq 0.83r_{\text{cur}}\zeta\lambda^-_{\text{cur}},$$

$$\lambda_{\max}(\boldsymbol{A}_\perp) \leq [\chi + 0.02r_{\text{cur}}\zeta]\lambda^-_{\text{cur}},$$

$$\lambda_{\max}(\boldsymbol{A}_\perp) \geq [\chi - 0.02r_{\text{cur}}\zeta]\lambda^-_{\text{cur}},$$

$$\lambda_{\min}(\boldsymbol{A}) \geq [1 - 0.0101r_{\text{cur}}\zeta]\lambda^-_{\text{cur}},$$

$$\lambda_{\max}(\boldsymbol{A}) \leq [g + 0.0202r_{\text{cur}}\zeta]\lambda^-_{\text{cur}},$$

$$\lambda_{\max}(\boldsymbol{A}) \geq [g - 0.02r_{\text{cur}}\zeta]\lambda^-_{\text{cur}}.$$

**Lemma 9.8.** *Consider the setting of Corollary 9.7. In addition, also assume that*

1. $\hat{g} = 1.01g + 0.0001$ *and*

2. $\chi \leq \min\left(\frac{g - 0.0001}{1.01g + 0.0001} - 0.0001, 1 - r_{\text{cur}}\zeta - \frac{0.08}{0.25}\right).$

*Let* $\hat{\lambda}_i := \lambda_i(\hat{\boldsymbol{D}})$. *Then, with probability at least* $1 - p_1 - p_2 - 4p_3$, *the following hold.*

1. *When* $\boldsymbol{G}_{\text{undet}}$ *is not empty:* $\frac{\hat{\lambda}_1}{\hat{\lambda}_{r_{\text{cur}}}} \leq \hat{g}$, $\frac{\hat{\lambda}_1}{\hat{\lambda}_{r_{\text{cur}}+1}} > \hat{g}$, *and* $\hat{\lambda}_{r_{\text{cur}}+1} \geq \lambda_{\text{thresh}}$.

2. *When* $\boldsymbol{G}_{\text{undet}}$ *is empty:* $\frac{\hat{\lambda}_1}{\hat{\lambda}_{r_{\text{cur}}}} \leq \hat{g}$ *and* $\hat{\lambda}_{r_{\text{cur}}+1} < \lambda_{\text{thresh}} < \hat{\lambda}_{r_{\text{cur}}}$.

3. *If* $\hat{r}_{\text{cur}} = r_{\text{cur}}$, *then* $\zeta_{\text{cur}} \leq \frac{\|\boldsymbol{H}\|}{\lambda_{\min}(\boldsymbol{A}) - \lambda_{\max}(\boldsymbol{A}_\perp) - \|\boldsymbol{H}\|} \leq 0.75r_{\text{cur}}\zeta + \frac{0.08r_{\text{cur}}\zeta}{(1 - r_{\text{cur}}\zeta - \chi)} \leq r_{\text{cur}}\zeta$.

*Proof.*

**Fact 9.9.** *From the bound on $\chi$, $\chi \leq 1 - 0.0001 \leq 1 - r_{\mathrm{cur}}\zeta$. Thus, using Corollary 9.7, $\lambda_{\min}(\boldsymbol{A}) > \lambda_{\max}(\boldsymbol{A}_\perp)$ and so $\lambda_{r_{\mathrm{cur}}}(\boldsymbol{D}) = \lambda_{\min}(\boldsymbol{A})$, $\lambda_{r_{\mathrm{cur}}+1}(\boldsymbol{D}) = \lambda_{\max}(\boldsymbol{A}_\perp)$, and $\lambda_1(\boldsymbol{D}) = \lambda_{\max}(\boldsymbol{A})$. Recall: $\lambda_1(.)$ is the same as $\lambda_{\max}(.)$.*

*Proof of item 1.* Recall that $\hat{\boldsymbol{D}}$ and $\boldsymbol{D}$ are defined in (10) and (11). Using Weyl's inequality, Fact 9.9, and Corollary 9.7, with the probability given there,

$$\frac{\hat{\lambda}_1}{\hat{\lambda}_{r_{\mathrm{cur}}}} \leq \frac{\lambda_{\max}(\boldsymbol{A}) + \|\boldsymbol{H}\|}{\lambda_{\min}(\boldsymbol{A}) - \|\boldsymbol{H}\|} \leq \frac{g + 0.86 r_{\mathrm{cur}}\zeta}{1 - 0.85 r_{\mathrm{cur}}\zeta}$$

and

$$\frac{\hat{\lambda}_1}{\hat{\lambda}_{r_{\mathrm{cur}}+1}} > \frac{\lambda_{\max}(\boldsymbol{A}) - \|\boldsymbol{H}\|}{\lambda_{\max}(\boldsymbol{A}_\perp) + \|\boldsymbol{H}\|} > \frac{g - 0.85 r_{\mathrm{cur}}\zeta}{\chi + 0.85 r_{\mathrm{cur}}\zeta}$$

Thus, if

$$\frac{g + 0.85 r_{\mathrm{cur}}\zeta}{1 - 0.85 r_{\mathrm{cur}}\zeta} \leq \hat{g} \leq \frac{g - 0.85 r_{\mathrm{cur}}\zeta}{\chi + 0.85 r_{\mathrm{cur}}\zeta} \tag{12}$$

holds, we will be done. The above requires $\chi$ to be small enough so that the lower bound is not larger than the upper bound and it requires $\hat{g}$ to be appropriately set. Both are ensured by the assumptions in the lemma.

Since $\boldsymbol{G}_{\mathrm{undet}}$ is not empty, $\lambda^+_{\mathrm{undet}} = \chi \lambda^-_{\mathrm{cur}} > \lambda^-$ Thus, using Weyl's inequality followed by Corollary 9.7, with the probability given there,

$$\begin{aligned}
\hat{\lambda}_{r_{\mathrm{cur}}+1} &\geq \lambda_{r_{\mathrm{cur}}+1}(\boldsymbol{D}) - \|\boldsymbol{H}\| = \lambda_{\max}(\boldsymbol{A}_\perp) - \|\boldsymbol{H}\| \\
&\geq [\chi - 0.02 r_{\mathrm{cur}}\zeta]\lambda^-_{\mathrm{cur}} - 0.83 r_{\mathrm{cur}}\zeta \lambda^-_{\mathrm{cur}} \\
&\geq (1 - 0.85 r_{\mathrm{cur}}\zeta)\lambda^- > \lambda_{\mathrm{thresh}}
\end{aligned}$$

*Proof of item 2.* Since $\boldsymbol{G}_{\mathrm{undet}}$ is empty, $\lambda^+_{\mathrm{undet}} = 0$ and so $\chi = 0$. Thus, using Corollary 9.7, with probability given there,

$$\begin{aligned}
\hat{\lambda}_{r_{\mathrm{cur}}+1} &\leq \lambda_{r_{\mathrm{cur}}+1}(\boldsymbol{D}) + \|\boldsymbol{H}\| = \lambda_{\max}(\boldsymbol{A}_\perp) + \|\boldsymbol{H}\| \\
&\leq 0 + 0.02 r_{\mathrm{cur}}\zeta \lambda^- + \|\boldsymbol{H}\| \leq 0.85 r_{\mathrm{cur}}\zeta \lambda^- \\
&< \lambda_{\mathrm{thresh}},
\end{aligned}$$

$$\begin{aligned}
\hat{\lambda}_{r_{\mathrm{cur}}} &\geq \lambda_{r_{\mathrm{cur}}}(\boldsymbol{D}) - \|\boldsymbol{H}\| = \lambda_{\min}(\boldsymbol{A}) - \|\boldsymbol{H}\| \\
&\geq \lambda^-_{\mathrm{cur}} - 0.085 r_{\mathrm{cur}}\zeta \lambda^-_{\mathrm{cur}} \geq (1 - 0.85 r_{\mathrm{cur}}\zeta)\lambda^- \\
&> \lambda_{\mathrm{thresh}},
\end{aligned}$$

and

$$\frac{\hat{\lambda}_1}{\hat{\lambda}_{r_{\mathrm{cur}}}} \leq \frac{\lambda_{\max}(\boldsymbol{A}) + \|\boldsymbol{H}\|}{\lambda_{\min}(\boldsymbol{A}) - \|\boldsymbol{H}\|} \leq \frac{g + 0.85 r_{\mathrm{cur}}\zeta}{1 - 0.85 r_{\mathrm{cur}}\zeta} \leq \hat{g}$$

*Proof of item 3.* Using Fact 9.5 and Corollary 9.7, since $\hat{r}_{\mathrm{cur}} = r_{\mathrm{cur}}$ is assumed, we get

$$\begin{aligned}
\zeta_{\mathrm{cur}} &\leq \frac{[0.75(1 - r_{\mathrm{cur}}\zeta - \chi)r_{\mathrm{cur}}\zeta + 0.08 r_{\mathrm{cur}}\zeta]\lambda^-_{\mathrm{cur}}}{\lambda^-_{\mathrm{cur}}[1 - 0.0101 r_{\mathrm{cur}}\zeta - \chi - 0.02 r_{\mathrm{cur}}\zeta - 0.83 r\zeta]} \\
&\leq \frac{0.75(1 - r\zeta - \chi)r_{\mathrm{cur}}\zeta + 0.08 r_{\mathrm{cur}}\zeta}{(1 - r_{\mathrm{cur}}\zeta - \chi)} \leq r_{\mathrm{cur}}\zeta \tag{13}
\end{aligned}$$

The last inequality used the bound on $\chi$. $\boxtimes$

## 9.3 Proof of Theorem 3.3

The theorem is a direct consequence of using (9) and applying Lemma 9.8 for each of the $k$ steps with the substitutions given in Definition 9.2; along with picking $\alpha$ appropriately. A detailed proof is in Sec. 11.

# 10 Proof of Hoeffding lemma, Lemma 9.6

The following lemma, which is a modification of [3, Lemma 8.15], will be used in our proof. It is proved in Sec. 11. The proof uses [3, Lemma 2.10].

**Lemma 10.1.** *Given* $\zeta_{\det} \leq r\zeta$.

1. $\|\boldsymbol{\Psi}\boldsymbol{G}_{\det}\| \leq r\zeta$ *and* $\|\boldsymbol{\Psi}\boldsymbol{G}_{\mathrm{cur}}\| \leq 1$.

2. $\sqrt{1-(r\zeta)^2} \leq \sigma_i(\boldsymbol{R}_{\mathrm{cur}}) = \sigma_i(\boldsymbol{\Psi}\boldsymbol{G}_{\mathrm{cur}}) \leq 1$ *and* $\sqrt{1-(r\zeta)^2} \leq \sigma_i(\boldsymbol{\Psi}\boldsymbol{G}_{\mathrm{undet}}) \leq 1$

3. $\|\boldsymbol{E}_{\mathrm{cur}}{}'\boldsymbol{\Psi}\boldsymbol{G}_{\mathrm{undet}}\| \leq \dfrac{(r\zeta)^2}{\sqrt{1-(r\zeta)^2}}$

4.
$$\boldsymbol{\Psi}\boldsymbol{\Sigma}\boldsymbol{\Psi} = [\boldsymbol{\Psi}\boldsymbol{G}_{\det} \ \boldsymbol{\Psi}\boldsymbol{G}_{\mathrm{cur}} \ \boldsymbol{\Psi}\boldsymbol{G}_{\mathrm{undet}}]$$
$$\begin{bmatrix} \boldsymbol{\Lambda}_{\det} & \mathbf{0} & \mathbf{0} \\ \mathbf{0} & \boldsymbol{\Lambda}_{\mathrm{cur}} & \\ \mathbf{0} & \mathbf{0} & \boldsymbol{\Lambda}_{\mathrm{undet}} \end{bmatrix} \begin{bmatrix} \boldsymbol{\Psi}\boldsymbol{G}_{\det} \\ \boldsymbol{\Psi}\boldsymbol{G}_{\mathrm{cur}} \\ \boldsymbol{\Psi}\boldsymbol{G}_{\mathrm{undet}} \end{bmatrix}'$$

    *with* $\lambda_{\max}(\boldsymbol{\Lambda}_{\det}) \leq \lambda^+$, $\lambda_{\mathrm{cur}}^- \leq \lambda_{\min}(\boldsymbol{\Lambda}_{\mathrm{cur}}) \leq \lambda_{\max}(\boldsymbol{\Lambda}_{\mathrm{cur}}) \leq \lambda_{\mathrm{cur}}^+$, $\lambda_{\max}(\boldsymbol{\Lambda}_{\mathrm{undet}}) \leq \lambda_{\mathrm{undet}}^+$.

5. *Using the first four claims, it is easy to see that*

    (a) $\|\boldsymbol{E}_{\mathrm{cur},\perp}{}'\boldsymbol{\Psi}\boldsymbol{\Sigma}\boldsymbol{\Psi}\boldsymbol{E}_{\mathrm{cur},\perp}\| \leq (r\zeta)^2\lambda^+ + \lambda_{\mathrm{undet}}^+$

    (b) $\|\boldsymbol{E}_{\mathrm{cur},\perp}{}'\boldsymbol{\Psi}\boldsymbol{\Sigma}\boldsymbol{\Psi}\boldsymbol{E}_{\mathrm{cur}}\| \leq (r\zeta)^2\lambda^+ + \dfrac{(r\zeta)^2}{\sqrt{1-(r\zeta)^2}}\lambda_{\mathrm{undet}}^+$

    (c) $\|\boldsymbol{\Psi}\boldsymbol{\Sigma}\| \leq (r\zeta)\lambda^+ + \lambda_{\mathrm{cur}}^+$ *and* $\|\boldsymbol{\Psi}\boldsymbol{\Sigma}\boldsymbol{M}_{1,t}{}'\| \leq q((r\zeta)\lambda^+ + \lambda_{\mathrm{cur}}^+)$

    (d) $\|\boldsymbol{M}_{1,t}\boldsymbol{\Sigma}\| \leq q\lambda^+$ *and* $\|\boldsymbol{M}_{1,t}\boldsymbol{\Sigma}\boldsymbol{M}_{1,t}{}'\| \leq q^2\lambda^+$

    *If* $\hat{\boldsymbol{G}}_{\det} = \boldsymbol{G}_{\det} = [.]$, *then all the terms containing* $(r\zeta)$ *disappear.*

6. $\lambda_{\min}(A + B) \geq \lambda_{\min}(A) + \lambda_{\min}(B)$

7. *Let* $\boldsymbol{a}_t := \boldsymbol{P}'\boldsymbol{\ell}_t$, $\boldsymbol{a}_{t,\det} := \boldsymbol{G}_{\det}{}'\boldsymbol{\ell}_t$, $\boldsymbol{a}_{t,\mathrm{cur}} := \boldsymbol{G}_{\mathrm{cur}}{}'\boldsymbol{\ell}_t$ *and* $\boldsymbol{a}_{t,\mathrm{undet}} := \boldsymbol{G}_{\mathrm{undet}}{}'\boldsymbol{\ell}_t$. *Also let* $\boldsymbol{a}_{t,\mathrm{rest}} := [\boldsymbol{a}_{t,\mathrm{cur}}{}', \boldsymbol{a}_{t,\mathrm{undet}}{}']'$. *Then* $\|\boldsymbol{a}_{t,\mathrm{rest}}\|^2 \leq r\eta\lambda_{\mathrm{cur}}^+$ *and* $\|\boldsymbol{a}_{t,\det}\|^2 \leq \|\boldsymbol{a}_t\|^2 \leq r\eta\lambda^+$.

8. $\sigma_{\min}(\boldsymbol{E}_{\mathrm{cur},\perp}{}'\boldsymbol{\Psi}\boldsymbol{G}_{\mathrm{undet}})^2 \geq 1 - (r\zeta)^2 - \dfrac{(r\zeta)^2}{\sqrt{1-(r\zeta)^2}}$.

The following corollaries of the matrix Hoeffding inequality [20], proved in [3], will be used in the proof.

**Corollary 10.2.** *Given an $\alpha$-length sequence $\{\boldsymbol{Z}_t\}$ of random Hermitian matrices of size $n \times n$, a r.v. $X$, and a set $\mathcal{C}$ of values that $X$ can take. For all $X \in \mathcal{C}$, (i) $\boldsymbol{Z}_t$'s are conditionally independent given $X$; (ii) $\mathbb{P}(b_1\boldsymbol{I} \preceq \boldsymbol{Z}_t \preceq b_2\boldsymbol{I}|X) = 1$ and (iii) $b_3\boldsymbol{I} \preceq \mathbb{E}[\frac{1}{\alpha}\sum_t \boldsymbol{Z}_t|X] \preceq b_4\boldsymbol{I}$. For any $\epsilon > 0$, for all $X \in \mathcal{C}$,*

$$\mathbb{P}\left(\lambda_{\max}\left(\frac{1}{\alpha}\sum_t \boldsymbol{Z}_t\right) \leq b_4 + \epsilon \Big| X\right) \geq 1 - n\exp\left(\frac{-\alpha\epsilon^2}{8(b_2-b_1)^2}\right),$$

$$\mathbb{P}\left(\lambda_{\min}\left(\frac{1}{\alpha}\sum_t \boldsymbol{Z}_t\right) \geq b_3 - \epsilon \Big| X\right) \geq 1 - n\exp\left(\frac{-\alpha\epsilon^2}{8(b_2-b_1)^2}\right).$$

**Corollary 10.3.** *Given an $\alpha$-length sequence $\{\boldsymbol{Z}_t\}$ of random matrices of size $n_1 \times n_2$. For all $X \in \mathcal{C}$, (i) $\boldsymbol{Z}_t$'s are conditionally independent given $X$; (ii) $\mathbb{P}(\|\boldsymbol{Z}_t\| \leq b_1|X) = 1$ and (iii) $\|\mathbb{E}[\frac{1}{\alpha}\sum_t \boldsymbol{Z}_t|X]\| \leq b_2$. For any $\epsilon > 0$, for all $X \in \mathcal{C}$,*

$$\mathbb{P}\left(\left\|\frac{1}{\alpha}\sum_t \boldsymbol{Z}_t\right\| \leq b_2 + \epsilon \Big| X\right) \geq 1 - (n_1 + n_2)\exp\left(\frac{-\alpha\epsilon^2}{32b_1{}^2}\right).$$

*Proof of Lemma 9.6.* Recall that we are given $\hat{\boldsymbol{G}}_{\det}$ that was computed using (some or all) $\boldsymbol{y}_t$'s for $t \leq t_*$ and that satisfies $\zeta_{\det} \leq r\zeta$. From (2), $\boldsymbol{y}_t$ is a linear function of $\boldsymbol{\ell}_t$. Thus, we can let $X := \{\boldsymbol{\ell}_1, \boldsymbol{\ell}_2, \dots \boldsymbol{\ell}_{t_*}\}$ denote all the random variables on which the event $\{\zeta_{\det} \leq r\zeta\}$ depends. In each item of this proof, we need to lower bound the probability of the desired event conditioned on $\zeta_{\det} \leq r\zeta$. To do this, we first lower bound the probability of the event conditioned on $X$ that is such that $X \in \{\zeta_{\det} \leq r\zeta\}$. We get a lower bound that does not depend on $X$ as long as $X \in \{\zeta_{\det} \leq r\zeta\}$. Thus, the same probability lower bound holds conditioned on $\{\zeta_{\det} \leq r\zeta\}$.

**Fact 10.4.** *For an event $\mathcal{E}$ and random variable $X$, $\mathbb{P}(\mathcal{E}|X) \geq p$ for all $X \in \mathcal{C}$ implies that $\mathbb{P}(\mathcal{E}|X \in \mathcal{C}) \geq p$.*

*Proof of Lemma 9.6, item 1.* Let

$$\text{term} := \frac{1}{\alpha} \sum_t \boldsymbol{\Psi}\boldsymbol{\ell}_t \boldsymbol{w}_t{}' = \frac{1}{\alpha} \sum_t \boldsymbol{\Psi}\boldsymbol{\ell}_t \boldsymbol{\ell}_t' \boldsymbol{M}_{1,t}{}' \boldsymbol{M}_{2,t}{}'$$

Since $\boldsymbol{\Psi}$ is a function of $X$, since $\boldsymbol{\ell}_t$'s used in the summation above are independent of $X$ and $\mathbb{E}[\boldsymbol{\ell}_t \boldsymbol{\ell}_t{}'] = \boldsymbol{\Sigma}$,

$$\mathbb{E}[\text{term}|X] = \frac{1}{\alpha} \sum_t \boldsymbol{\Psi}\boldsymbol{\Sigma}\boldsymbol{M}_{1,t}{}' \boldsymbol{M}_{2,t}{}'$$

Next, we use Cauchy-Schwartz for matrices:

$$\left\| \sum_{t=1}^{\alpha} \boldsymbol{X}_t \boldsymbol{Y}_t{}' \right\|^2 \leq \lambda_{\max}\left( \sum_{t=1}^{\alpha} \boldsymbol{X}_t \boldsymbol{X}_t{}' \right) \lambda_{\max}\left( \sum_{t=1}^{\alpha} \boldsymbol{Y}_t \boldsymbol{Y}_t{}' \right) \tag{14}$$

Using (14), with $\boldsymbol{X}_t = \boldsymbol{\Psi}\boldsymbol{\Sigma}\boldsymbol{M}_{1,t}{}'$ and $\boldsymbol{Y}_t = \boldsymbol{M}_{2,t}$, followed by using $\sqrt{\|\frac{1}{\alpha}\sum_t \boldsymbol{X}_t \boldsymbol{X}_t'\|} \leq \max_t \|\boldsymbol{X}_t\|$, Assumption 1.2 with $\boldsymbol{A}_t \equiv \boldsymbol{I}$, and Lemma 10.1,

$$\|\mathbb{E}[\text{term}|X]\| \leq \max_t \|\boldsymbol{\Psi}\boldsymbol{\Sigma}\boldsymbol{M}_{1,t}{}'\| \sqrt{\frac{\beta}{\alpha}}$$

$$\leq q((r\zeta)\lambda^+ + \lambda_{\text{cur}}^+)\sqrt{\frac{\beta}{\alpha}}$$

for all $X \in \{\zeta_{\det} \leq r\zeta\}$. To bound $\|\boldsymbol{\Psi}\boldsymbol{\ell}_t \boldsymbol{w}_t'\|$, rewrite it as $\boldsymbol{\Psi}\boldsymbol{\ell}_t \boldsymbol{w}_t' = [\boldsymbol{\Psi}\boldsymbol{G}_{\det}a_{t,\det} + \boldsymbol{\Psi}\boldsymbol{G}_{\text{rest}}a_{t,\text{rest}}][a_{t,\det}'\boldsymbol{G}_{\det}' + a_{t,\text{rest}}'\boldsymbol{G}_{\text{rest}}']\boldsymbol{M}_{1,t}'\boldsymbol{M}_{2,t}'$. Thus, using $\|\boldsymbol{M}_{2,t}\| \leq 1$, $\|\boldsymbol{M}_{1,t}\boldsymbol{P}\| \leq q < 1$, and Lemma 10.1,

$$\|\boldsymbol{\Psi}\boldsymbol{\ell}_t \boldsymbol{w}_t'\| \leq qr\eta((r\zeta)\lambda^+ + \lambda_{\text{cur}}^+ + (r\zeta)\sqrt{\lambda^+\lambda_{\text{cur}}^+} + \sqrt{\lambda^+\lambda_{\text{cur}}^+})$$

holds w.p. one when $\{\zeta_{\det} \leq r\zeta\}$.

Finally, conditioned on $X$, the individual summands in $\text{term}$ are conditionally independent. Using matrix Hoeffding, Corollary 10.3, followed by Fact 10.4, the result follows.

*Proof of Lemma 9.6, item 2.*

$$\mathbb{E}[\frac{1}{\alpha} \sum_t \boldsymbol{w}_t \boldsymbol{w}_t'|X] = \frac{1}{\alpha} \sum_t \boldsymbol{M}_{2,t}\boldsymbol{M}_{1,t}\boldsymbol{\Sigma}\boldsymbol{M}_{1,t}{}'\boldsymbol{M}_{2,t}{}'$$

By Lemma 10.1, $\|\boldsymbol{M}_{1,t}\boldsymbol{\Sigma}\boldsymbol{M}_{1,t}{}'\| \leq q^2\lambda^+$. Thus, using Assumption 1.2 with $\boldsymbol{A}_t \equiv \boldsymbol{M}_{1,t}\boldsymbol{\Sigma}\boldsymbol{M}_{1,t}{}'$,

$$\|\mathbb{E}[\frac{1}{\alpha} \sum_t \boldsymbol{w}_t \boldsymbol{w}_t'|X]\| \leq \frac{\beta}{\alpha}q^2\lambda^+.$$

Using Assumption 1.2 and Lemma 10.1,

$$\|\boldsymbol{w}_t \boldsymbol{w}_t'\| = \|\boldsymbol{M}_{2,t}\boldsymbol{M}_{1,t}\boldsymbol{P}a_t\|^2 \leq q^2\eta r\lambda^+.$$

Conditional independence of the summands holds as before. Thus, using Corollary 10.3 and Fact 10.4, the result follows.

*Proof of Lemma 9.6, item 3.*

$$\mathbb{E}[\frac{1}{\alpha}\sum_t \boldsymbol{E}_{\mathrm{cur}}\boldsymbol{E}_{\mathrm{cur}}{}'\boldsymbol{\Psi}\boldsymbol{\ell}_t\boldsymbol{\ell}_t{}'\boldsymbol{\Psi}\boldsymbol{E}_{\mathrm{cur},\perp}\boldsymbol{E}_{\mathrm{cur},\perp}{}'\||X]$$
$$= \boldsymbol{E}_{\mathrm{cur}}\boldsymbol{E}_{\mathrm{cur}}{}'\boldsymbol{\Psi}\boldsymbol{\Sigma}\boldsymbol{\Psi}\boldsymbol{E}_{\mathrm{cur},\perp}\boldsymbol{E}_{\mathrm{cur},\perp}{}'$$

Using Lemma 10.1, $\|\boldsymbol{E}_{\mathrm{cur}}\boldsymbol{E}_{\mathrm{cur}}{}'\boldsymbol{\Psi}\boldsymbol{\Sigma}\boldsymbol{\Psi}\boldsymbol{E}_{\mathrm{cur},\perp}\boldsymbol{E}_{\mathrm{cur},\perp}{}'\| \leq (r\zeta)^2\lambda^+ + \frac{(r\zeta)^2}{\sqrt{1-(r\zeta)^2}}\lambda^+_{\mathrm{undet}}$ when $\{\zeta_{\det} \leq r\zeta\}$. Also, $\|\boldsymbol{E}_{\mathrm{cur}}{}'\boldsymbol{\Psi}\boldsymbol{\ell}_t\boldsymbol{\ell}_t{}'\boldsymbol{\Psi}\boldsymbol{E}_{\mathrm{cur},\perp}\| \leq \|\boldsymbol{\Psi}\boldsymbol{\ell}_t\boldsymbol{\ell}_t{}'\boldsymbol{\Psi}\| \leq \eta r((r\zeta)^2\lambda^+ + \lambda^+_{\mathrm{cur}} + 2(r\zeta)\sqrt{\lambda^+\lambda^+_{\mathrm{cur}}}) := b_{prob}$ holds w.p. one when $\{\zeta_{\det} \leq r\zeta\}$. In the above bound, the first inequality is used to get a loose bound, but one that will also apply for the proofs of the later items given below. The rest is the same as in the proofs of the earlier parts.

*Proof of Lemma 9.6, item 4.* Using Ostrowski's theorem,

$$\lambda_{\min}(\mathbb{E}[\boldsymbol{A}|X]) = \lambda_{\min}(\boldsymbol{E}_{\mathrm{cur}}{}'\boldsymbol{\Psi}(\boldsymbol{\Sigma})\boldsymbol{\Psi}\boldsymbol{E}_{\mathrm{cur}})$$
$$\geq \lambda_{\min}(\boldsymbol{E}_{\mathrm{cur}}{}'\boldsymbol{\Psi}\boldsymbol{G}_{\mathrm{cur}}\boldsymbol{\Lambda}_{\mathrm{cur}}\boldsymbol{G}_c ur'\boldsymbol{\Psi}\boldsymbol{E}_{\mathrm{cur}})$$
$$= \lambda_{\min}(\boldsymbol{R}_{\mathrm{cur}}\boldsymbol{\Lambda}_{\mathrm{cur}}\boldsymbol{R}_{\mathrm{cur}}{}')$$
$$\geq \lambda_{\min}(\boldsymbol{R}_{\mathrm{cur}}\boldsymbol{R}_{\mathrm{cur}}{}')\lambda_{\min}(\boldsymbol{\Lambda}_{\mathrm{cur}}) \geq (1-(r\zeta)^2)\lambda^-_{\mathrm{cur}}$$

for all $X \in \{\zeta_{\det} \leq r\zeta\}$. Ostrowski's theorem is used to get the second-last inequality, while Lemma 10.1 helps get the last one.

As in the proof of item 3, $\|\boldsymbol{E}_{\mathrm{cur}}{}'\boldsymbol{\Psi}\boldsymbol{\ell}_t\boldsymbol{\ell}_t{}'\boldsymbol{\Psi}\boldsymbol{E}_{\mathrm{cur}}\| \leq \|\boldsymbol{\Psi}\boldsymbol{\ell}_t\boldsymbol{\ell}_t{}'\boldsymbol{\Psi}\| \leq b_{prob}$ holds w.p. one when $\{\zeta_{\det} \leq r\zeta\}$. Conditional independence of the summands holds as before. Thus, by matrix Hoeffding, Corollary 10.2, the result follows.

*Proof of Lemma 9.6, item 5.* By Lemma 10.1,

$$\lambda_{\max}(\mathbb{E}[\boldsymbol{A}_\perp|X]) = \lambda_{\max}(\boldsymbol{E}_{\mathrm{cur},\perp}{}'\boldsymbol{\Psi}\boldsymbol{\Sigma}\boldsymbol{\Psi}\boldsymbol{E}_{\mathrm{cur},\perp})$$
$$\leq ((r\zeta)^2\lambda^+ + \lambda^+_{\mathrm{undet}})$$

when $\{\zeta_{\det} \leq r\zeta\}$. The rest of the proof is the same as that of the previous part.

*Proof of Lemma 9.6, item 6.* Using Ostrowski's theorem, $\lambda_{\max}(\mathbb{E}[\boldsymbol{A}_\perp|X]) \geq \lambda_{\max}(\boldsymbol{E}_{\mathrm{cur},\perp}{}'\boldsymbol{\Psi}\boldsymbol{G}_{\mathrm{undet}}\boldsymbol{\Lambda}_{\mathrm{undet}}\boldsymbol{G}_{\mathrm{undet}}{}'\boldsymbol{\Psi}\boldsymbol{E}_{\mathrm{cur},\perp}) \geq \lambda_{\min}(\boldsymbol{E}_{\mathrm{cur},\perp}{}'\boldsymbol{\Psi}\boldsymbol{G}_{\mathrm{undet}}\boldsymbol{G}_{\mathrm{undet}}{}'\boldsymbol{\Psi}\boldsymbol{E}_{\mathrm{cur},\perp})\lambda_{\max}(\boldsymbol{\Lambda}_{\mathrm{undet}})$. By definition, $\lambda_{\max}(\boldsymbol{\Lambda}_{\mathrm{undet}}) = \lambda^+_{\mathrm{undet}}$. By Lemma 10.1, $\lambda_{\min}(\boldsymbol{E}_{\mathrm{cur},\perp}{}'\boldsymbol{\Psi}\boldsymbol{G}_{\mathrm{undet}}\boldsymbol{G}_{\mathrm{undet}}{}'\boldsymbol{\Psi}\boldsymbol{E}_{\mathrm{cur},\perp}) = \sigma_{\min}(\boldsymbol{E}_{\mathrm{cur},\perp}{}'\boldsymbol{\Psi}\boldsymbol{G}_{\mathrm{undet}})^2 \geq (1-(r\zeta)^2 - \frac{(r\zeta)^2}{\sqrt{1-(r\zeta)^2}})$ when $\{\zeta_{\det} \leq r\zeta\}$. The rest of the proof is the same as above.

*Proof of Lemma 9.6, item 7.* Using Ostrowski's theorem and Lemma 10.1, $\lambda_{\max}(\mathbb{E}[\boldsymbol{A}|X]) \geq \lambda_{\max}(\boldsymbol{E}_{\mathrm{cur}}{}'\boldsymbol{\Psi}\boldsymbol{G}_{\mathrm{cur}}\boldsymbol{\Lambda}_{\mathrm{cur}}\boldsymbol{G}_{\mathrm{cur}}{}'\boldsymbol{\Psi}\boldsymbol{E}_{\mathrm{cur}}) \geq \lambda_{\min}(\boldsymbol{R}_{\mathrm{cur}}\boldsymbol{R}_{\mathrm{cur}}{}')\lambda_{\max}(\boldsymbol{\Lambda}_{\mathrm{cur}}) \geq (1-(r\zeta)^2)\lambda^+_{\mathrm{cur}}$ when $\{\zeta_{\det} \leq r\zeta\}$. The rest of the proof is the same as above. $\boxtimes$

## 11 Detailed Proof of Theorem 3.3 and Proof of Lemma 10.1

*Proof of Theorem 3.3.* Recall that we need to show that $\zeta_k \leq r_k\zeta$. Assume the substitutions given in Definition 9.2. We will use induction.

Consider a $k < \vartheta$. For the $k$-th step, assume that $\zeta_i \leq r_i\zeta$ for $i = 1, 2, \ldots, k-1$. Thus, using (9), $\zeta_{\det} \leq r\zeta$ and so Lemma 9.8 is applicable. We first show that $\hat{r}_k = r_k$ and that Algorithm 1 does not stop (proceeds to $(k+1)$-th step). From Algorithm 1, $\hat{r}_k = r_k$ if $\frac{\hat{\lambda}_1}{\hat{\lambda}_{r_k}} \leq \hat{g}$, and $\frac{\hat{\lambda}_1}{\hat{\lambda}_{r_k+1}} > \hat{g}$. Also it will not stop if $\hat{\lambda}_{r_k+1} \geq \lambda_{\mathrm{thresh}}$. Since $k < \vartheta$, $\boldsymbol{G}_{\mathrm{undet}}$ is not empty. Thus, item 1 of Lemma 9.8 shows that all these hold. Hence $\hat{r}_k = r_k$ and algorithm does not stop w.p. at least $1 - p_1 - p_2 - 4p_3$. Thus, by item 3 of the same lemma, with the same probability, $\zeta_k \leq r_k\zeta$.

Now consider $k = \vartheta$. We first show $\hat{r}_k = r_k$ and that Algorithm 1 does stop, i.e., $\hat{\vartheta} = \vartheta$. This will be true if $\frac{\hat{\lambda}_1}{\hat{\lambda}_{r_k}} \leq \hat{g}$ and $\hat{\lambda}_{r_k+1} < \lambda_{\mathrm{thresh}}$. For $k = \vartheta$, $\boldsymbol{G}_{\mathrm{undet}}$ is empty. Thus, item 2 of Lemma 9.8 shows that this holds w.p. at least $1 - p_1 - p_2 - 4p_3$. Thus, by item 3 of the same lemma, with the same probability, $\zeta_k \leq r_k\zeta$.

Thus, using the union bound, w.p. at least $1 - \vartheta(p_1 + p_2 + 4p_3)$, $\hat{r}_k = r_k$ and $\zeta_k \leq r_k\zeta$ for all $k$. Using (8), this implies that $\text{SE} \leq r\zeta$ with the same probability.

Finally, the choice $\alpha \geq \alpha_0$, implies that $p_1 \leq \frac{1}{\vartheta}2n^{-10}$, $p_2 \leq \frac{1}{\vartheta}2n^{-10}$, $p_3 \leq \frac{1}{\vartheta}2n^{-10}$. Hence $\text{SE} \leq r\zeta$ w.p. at least $1 - 12n^{-10}$. We work this out for $p_1$ below. The others follow similarly.

Recall that $p_1 = 2n\exp(-\alpha\frac{\epsilon^2}{32b_{prob}^2})$, $\epsilon = 0.01(r\zeta)\lambda^-$ and $b_{prob} = \eta r q((r\zeta)\lambda^+ + \lambda_{\text{cur}}^+ + (r\zeta)\sqrt{\lambda^+\lambda_{\text{cur}}^+} + \sqrt{\lambda^+\lambda_{\text{cur}}^+})$. Thus, $\frac{b_{prob}^2}{(\lambda^-)^2} \leq (4\eta r \max(q(r\zeta)f, qg, q\sqrt{fg}, q(r\zeta)\sqrt{fg}))^2 \leq 16\eta^2 r^2 \max(q(r\zeta)f, qg, q\sqrt{fg})^2$

Thus, $\alpha\frac{\epsilon^2}{32b_{prob}^2} \geq \frac{32\cdot16}{(0.01)^2}\frac{\eta^2 r^2(11\log n + \log\vartheta)}{(r\zeta)^2}\max(q(r\zeta)f, qg, q\sqrt{fg})\frac{(0.01(r\zeta))^2}{32\cdot16\eta^2 r^2 \max(q(r\zeta)f, qg, q\sqrt{fg})^2} \geq 11\log n + \log\vartheta$. Thus, $p_1 \leq \frac{1}{\vartheta}2n^{-10}$. $\boxtimes$

*Proof of Lemma 10.1.* The first claim is obvious. The next two claims follow using the following lemma:

**Lemma 11.1** ([3], Lemma 2.10). *Suppose that $\boldsymbol{P}$, $\hat{\boldsymbol{P}}$ and $\boldsymbol{Q}$ are three basis matrices. Also, $\boldsymbol{P}$ and $\hat{\boldsymbol{P}}$ are of the same size, $\boldsymbol{Q}'\boldsymbol{P} = 0$ and $\|(\boldsymbol{I} - \hat{\boldsymbol{P}}\hat{\boldsymbol{P}}')\boldsymbol{P}\| = \zeta_*$. Then,*

1. *$\|(\boldsymbol{I} - \hat{\boldsymbol{P}}\hat{\boldsymbol{P}}')\boldsymbol{P}\boldsymbol{P}'\| = \|(\boldsymbol{I} - \boldsymbol{P}\boldsymbol{P}')\hat{\boldsymbol{P}}\hat{\boldsymbol{P}}'\| = \|(\boldsymbol{I} - \boldsymbol{P}\boldsymbol{P}')\hat{\boldsymbol{P}}\| = \|(\boldsymbol{I} - \hat{\boldsymbol{P}}\hat{\boldsymbol{P}}')\boldsymbol{P}\| = \zeta_*$*

2. *$\|\boldsymbol{P}\boldsymbol{P}' - \hat{\boldsymbol{P}}\hat{\boldsymbol{P}}'\| \leq 2\|(\boldsymbol{I} - \hat{\boldsymbol{P}}\hat{\boldsymbol{P}}')\boldsymbol{P}\| = 2\zeta_*$*

3. *$\|\hat{\boldsymbol{P}}'\boldsymbol{Q}\| \leq \zeta_*$*

4. *$\sqrt{1 - \zeta_*^2} \leq \sigma_i\left((\boldsymbol{I} - \hat{\boldsymbol{P}}\hat{\boldsymbol{P}}')\boldsymbol{Q}\right) \leq 1$*

Use item 4 of Lemma 11.1 and the fact that $\boldsymbol{G}_{\det}'\boldsymbol{G}_{\text{cur}} = \boldsymbol{0}$ and $\boldsymbol{G}_{\det}'\boldsymbol{G}_{\text{undet}} = \boldsymbol{0}$ to get the second claim.

For the third claim, notice that $\boldsymbol{E}_{\text{cur}}'\boldsymbol{\Psi}\boldsymbol{G}_{\text{undet}} = \boldsymbol{R}_{\text{cur}}^{-1}\boldsymbol{G}_{\text{cur}}'\boldsymbol{\Psi}\boldsymbol{G}_{\text{undet}} = \boldsymbol{R}_{\text{cur}}^{-1}\boldsymbol{G}_{\text{cur}}'\hat{\boldsymbol{G}}_{\det}\hat{\boldsymbol{G}}_{\det}'\boldsymbol{G}_{\text{undet}}$. since $\boldsymbol{\Psi}^2 = \boldsymbol{\Psi}$ and $\boldsymbol{G}_{\text{cur}}'\boldsymbol{G}_{\text{undet}} = \boldsymbol{0}$. Using the second claim, $\|\boldsymbol{R}_{\text{cur}}^{-1}\| \leq \frac{1}{\sigma_{\min}(\boldsymbol{R}_{\text{cur}})} \leq \frac{1}{1-(r\zeta)^2}$. Use item 3 of Lemma 11.1 and the facts that $\boldsymbol{G}_{\text{cur}}'\boldsymbol{G}_{\det} = \boldsymbol{0}$ and $\boldsymbol{G}_{\text{undet}}'\boldsymbol{G}_{\det} = \boldsymbol{0}$ to bound $\|\boldsymbol{G}_{\text{cur}}'\hat{\boldsymbol{G}}_{\det}\|$ and $\|\hat{\boldsymbol{G}}_{\det}'\boldsymbol{G}_{\text{undet}}\|$ respectively.

The fourth claim just uses the definitions. The fifth claim uses the previous claims and the assumptions on $\boldsymbol{M}_t$ from Assumption 1.2. The sixth claim follows using Weyl's inequality.

The second last claim: We show how to bound $\boldsymbol{a}_{t,\text{rest}}$: $\|a_{t,\text{rest}}\|^2 = \|a_{t,\text{cur}}\|^2 + \|a_{t,\text{undet}}\|^2 \leq \sum_{j\in\mathcal{G}_{\text{cur}}}\eta\lambda_j + \sum_{j\in\mathcal{G}_{\text{undet}}}\eta\lambda_j \leq r\eta\lambda_{\text{cur}}^+$ (since $\lambda_j \leq \lambda_{\text{cur}}^+$ for all the $j$'s being summed over). The other bounds follow similarly.

Last claim:

$$\sigma_{\min}(\boldsymbol{E}_{\text{cur},\perp}'\boldsymbol{\Psi}\boldsymbol{G}_{\text{undet}})^2$$
$$= \lambda_{\min}(\boldsymbol{G}_{\text{undet}}'\boldsymbol{\Psi}\boldsymbol{E}_{\text{cur},\perp}\boldsymbol{E}_{\text{cur},\perp}'\boldsymbol{\Psi}\boldsymbol{G}_{\text{undet}})$$
$$= \lambda_{\min}(\boldsymbol{G}_{\text{undet}}'\boldsymbol{\Psi}(\boldsymbol{I} - \boldsymbol{E}_{\text{cur}}\boldsymbol{E}_{\text{cur}}')\boldsymbol{\Psi}\boldsymbol{G}_{\text{undet}})$$
$$\geq \lambda_{\min}(\boldsymbol{G}_{\text{undet}}'\boldsymbol{\Psi}\boldsymbol{\Psi}\boldsymbol{G}_{\text{undet}})-$$
$$\lambda_{\max}(\boldsymbol{G}_{\text{undet}}'\boldsymbol{\Psi}\boldsymbol{E}_{\text{cur}}\boldsymbol{E}_{\text{cur}}'\boldsymbol{\Psi}\boldsymbol{G}_{\text{undet}})$$
$$= \sigma_{\min}(\boldsymbol{\Psi}\boldsymbol{G}_{\text{undet}})^2 - \|\boldsymbol{E}_{\text{cur}}'\boldsymbol{\Psi}\boldsymbol{G}_{\text{undet}}\|$$
$$\geq 1 - (r\zeta)^2 - \frac{(r\zeta)^2}{\sqrt{1-(r\zeta)^2}}.$$

The last inequality follows using the second and the third claim. $\boxtimes$

## Footnotes

[1]This follows easily by writing $\boldsymbol{H} = (\hat{\boldsymbol{D}} - \frac{1}{\alpha} \sum_t \boldsymbol{\Psi} \boldsymbol{\ell}_t \boldsymbol{\ell}_t' \boldsymbol{\Psi}) + (\frac{1}{\alpha} \sum_t \boldsymbol{\Psi} \boldsymbol{\ell}_t \boldsymbol{\ell}_t' \boldsymbol{\Psi} - \boldsymbol{D})$ and using the fact that $\boldsymbol{M} = (\boldsymbol{E} \boldsymbol{E}' + \boldsymbol{E}_\perp \boldsymbol{E}_\perp') \boldsymbol{M} (\boldsymbol{E} \boldsymbol{E}' + \boldsymbol{E}_\perp \boldsymbol{E}_\perp')$ for $\frac{1}{\alpha} \sum_t \boldsymbol{\Psi} \boldsymbol{\ell}_t \boldsymbol{\ell}_t' \boldsymbol{\Psi}$.