[Reviews · NeurIPS 2016]

Reviewer 1

Summary

The paper analyzes the PCA (eigen value decomposition -EVD) for the cases where underlying data and noise are correlated. The authors prove the correctness of this approach, providing an upper bound on the subspace recovery error, and then provide a generalization of PCA - cluster EVD, which overcomes limitations on the sample complexity of PCA, which is bound by the condition number of the covariance matrix. The paper provides various models of correlation between data and noise, and provides background literature on the recovery error and sample complexity for existing approaches that solve this problem. The main contribution is the derivation of cluster-EVD, and proving the upperbound on the sample complexity when assumptions on the nature of clustering of eigen values across scales holds.

Qualitative Assessment

The paper analyzes the PCA (eigen value decomposition -EVD) for the cases where underlying data and noise are correlated. The authors prove the correctness of this approach, providing an upper bound on the subspace recovery error, and then provide a generalization of PCA - cluster EVD, which overcomes limitations on the sample complexity of PCA, which is bound by the condition number of the covariance matrix. The adaptation of clustering of eigen vlaues and the strategy used are novel. The proofs are reasonably well structured and clear. The authors could possibly position theoram 3.3 earlier in the text, and add the proof outline later. The authors study some cases of videos where the assumption of clustered eigen values is observed. Some more practical scenarios will be useful to understand the scope of applicability of the algorithm. The experiment section is validates the method in their chosen metrics, in comparison to other methods that solve similar problems in PCA. The authors clarify that there are certain generalized cases (where the clustered eigenvalues assumption doesn't hold, assumption on missing/corrupted values doesn't hold), where cluster-EVD is not optimal. A possible experiment would be a comparison with other methods when these assumptions are not satisfied. A satisfactory performance would be show that cluster-EVD can be used in less restrictive cases as well. Overall, the paper provides a key observation that solves a specific problem. The paper is well written.

Confidence in this Review

2-Confident (read it all; understood it all reasonably well)


Reviewer 2

Summary

The paper provides probabilistic error bounds on eigenvectors if the data are distortioned in a very specific data dependent way.

Qualitative Assessment

Avoid in the title and the abstract the use of terminology that also has a specific meaning in statistics, different of what the authors have in mind (e.g. noise, correlation, etc). Show with some specific examples the intuition and meaning behind restriction (2). Inclusion of real data sets might be clarifying also to see how to check (2)

Confidence in this Review

1-Less confident (might not have understood significant parts)


Reviewer 3

Summary

This paper tackles the problem of principal component analysis when the data and noise are correlated. The sample complexity under standard principal component analysis via singular/eigen value decomposition is shown to depend on the square of the condition number of the true covariance matrix of the data which can be large when the data and error are correlated. The authors propose a cluster eigen value decomposition (EVD) under the assumption that the eigenvalues of the true covariance matrix is clustered. Under the cluster EVD paradigm, the sample complexity depends on the square of the condition number within each cluster.

Qualitative Assessment

The paper tackles an interesting problem. Correlated PCA seems to be a problem that has not been addressed yet in current literature in complete generality even though some subproblems of it has been addressed. However, I found the presentation of the paper not very lucid, especially for an uninitiated researcher. Several theoretical assumptions and comments are made without much explanation. I have the following specific comments and suggestions: 1. What is the implication of the assumptions in Model 1.2? Is it a standard assumption for problems of this kind? This assumption is fairly technical; should it be presented as part of the main part of the paper? 2. In theorem 2.1, r is referenced but not defined prior to it (as far as I saw). I believe it is the rank of the true covariance matrix? 3. In model 3.1, the "well-separation" is defined as the ratio of largest eigen value of (k+1)-th cluster and smallest eigen value in k-th cluster being upper-bounded by some constant chi. Could the authors provide an intuition for this definition? Should it not be lower bounded instead? Some more explanation would be very helpful. 4. The authors provide examples where cluster assumptions do hold. How does this procedure work in cases when the assumption is violated?

Confidence in this Review

2-Confident (read it all; understood it all reasonably well)


Reviewer 4

Summary

The paper is concerned with the formulation of a theory and algorithms for optimal recovery of a low rank subspace in case that noise and data are correlated. This area of research is interesting and understudied.

Qualitative Assessment

My main concern regarding the methodologies proposed in the paper is that they are special cases or trivial reformulations of the theory developed in [18]. For example the main algorithm of the paper (Algorithm 1 Cluster EVD) is a special case of Algorithm 2 (Cluster PCA) of [18]. Furthermore all the theory, models, assumptions of the paper (i.e., section 1) are taken directly from [18] (even the wording is taken from [18]). I would like the authors to describe in as much detail as possible what are the differences and advances over 18.

Confidence in this Review

2-Confident (read it all; understood it all reasonably well)


Reviewer 5

Summary

The paper introduces a novel derivation of Principal Component Analysis (PCA) that introduces correlation between the signal and the noise. It also introduces a refinement of the classical eigenvalue decomposition (EVD) that assumes that the eigenvalues are clustered; this can help reduce the complexity of EVD and, therefore, that of the proposed PCA method.

Qualitative Assessment

The large number of assumptions scattered throughout the paper (although some of them seem reasonable) does raise some concerns about the usability of the method on real data. An experiment with real data (or even some other simulated experiment) would considerably improve the paper and alleviate any possible concern about that.

Confidence in this Review

2-Confident (read it all; understood it all reasonably well)


Reviewer 6

Summary

This paper deals with a similar but different problem setting for principal component analysis (PCA) where the (uncontaminated) true data vector and noise vector are correlated. Under this setting, the sample complexity of the conventional PCA does not scale with the condition number of the data covariance matrix. To cope with this problem, the authors proposed an algorithm called cluster-EVD. According to theoretical analysis, cluser-EVD scales with the conditional number much more mildly.

Qualitative Assessment

This paper tackles with a PCA-like subspace recovery problem under an interesting problem setting where the true data vector and noise vector are correlated. To cope with a problem of the conventional PCA under this setting, the authors proposed a new algorithm called cluster-EVD. Some property of cluster-EVD is theoretically investigated. However, this paper includes some significant problems. The main drawback of this paper is that it is unclear whether the proposed method is really useful in practice. The experimental results are very limited. Furthermore, Table 1 shows that cluster-EVD and EVD show almost same performance. Therefore, the result is not convincing. In addition, there are a couple of parameters such as $\alpha$ and $\hat{g}$ in Algorithm 1. How do you tune these parameters? Is there any principled way to tune these parameters? Thus, I am skeptical about the practical usefulness of the proposed method. Another drawback to me is that cluster-EVD is not justified well. For example, from line 160 to 171, they presented some motivating examples for cluster-EVD, and showed a couple of parameter values such as $g$, $\vartheta$ and $\chi$. However, I am not sure that these parameter values are good enough to justify that cluster-EVD covers typical practical situations.

Confidence in this Review

2-Confident (read it all; understood it all reasonably well)